

**The full title of the paper**
Trawl macrofauna of the Far-Eastern Seas and North Pacific: proportion of commercial species,
potential product yield, and price range
**A suggested short running title**
Commercial value of the North Pacific trawl macrofauna
**The full names of all authors**
*Igor V. Volvenko[1], Alexei M. Orlov[2,3,4,5,6], Andrey V. Gebruk[7], Oleg N. Katugin[1], Alla A.*
*Ogorodnikova[1], Georgy M. Vinogradov[7], Olga A. Maznikova[2]*
**The author's institutional affiliations**
[1]Pacific Branch of Russian Federal Research Institute of Fisheries and Oceanography (TINRO),
Vladivostok, 690091 Russia; [2]Russian Federal Research Institute of Fisheries and Oceanography
(VNIRO), Moscow, 107140 Russia; [3]A.N. Severtsov Institute of Ecology and Evolution, Russian
Academy of Sciences (IPEE), Moscow, 119071 Russia; [4]Dagestan State University (DSU),
Makhachkala, 367000 Russia; [5]Tomsk State University (TSU), Tomsk, 634050 Russia; [6]Caspian
Institute of Biological Resources, Dagestan Scientific Center of the Russian Academy of
Sciences (CIBR), Makhachkala, 367000 Russia; [7]Shirshov Institute of Oceanology, Russian
Academy of Sciences (IO RAS), Moscow, 117997 Russia
**Correspondence**
Igor V. Volvenko, TINRO, 4 Shevchenko Ave., Vladivostok, 690091, Russian Federation, E-
mail: oknevlov@gmail.com, ORCiD ID: 0000-0003-0369-4039



**Abstract**
A checklist of 1541 animal species from the Chukchi, Bering, Okhotsk, and Japan seas
and the North Pacific Ocean was generated based on 459 research vessel surveys (68903 trawl
tows at depths from 5 to 2200 m) in the period 1977-2014 (Volvenko et al.,
https://doi.pangaea.de/10.1594/PANGAEA.902195, 2019). The study area spanned over 25 million km$^2$.
For each species, the scientific name is given, as well as English and Russian common names
along with the following details: areas where species were collected, trawl type
(benthic/midwater), real or potential commercial importance, possible product yield and
minimum wholesale prices. Almost 20% of species in trawl catches had no commercial value,
and >50% were cheap or very cheap. Only 3.3% of species were expensive and very expensive,
and their number increased from north to south. About 33% of species can be considered as
unexploited reserve for fisheries. These are mainly small fish and invertebrates, with total
biomass many times exceeding that of currently exploited biological resources. Product output
for most species exceeded 90% of the raw weight. Occurrence of such species was much higher
in the pelagic zone than on the seafloor. The most abundant local commercial species are
characterized by significant natural fluctuations in abundance. Therefore, a sustainable fishery in
the region can only be secured by expansion of the assortment of commercial bioresources. A
regional supply of bioresources provides such an opportunity. The checklist can be used for
development of bioresource management, aquaculture and conservation, assessment of
environmental damage caused by anthropogenic impact (hydro-technical constructions, oil/gas
extractions, nuclear reactor accidents, etc.).
**Keywords**
Commercial importance, comparison of marine basins, North Pacific and East Arctic, product
yield and prices, species checklist, trawl catches





**Sections**



## Introduction


The region where material for the present study was collected (Fig. 1) is one of the most

productive and economically important regions in the World Ocean (Zenkevich, 1963; Moiseev,
1969; Bogorov, 1970; Gershanovich et al., 1990; Shuntov, 2001, 2016). It includes the Chukchi
and Bering seas, Sea of Okhotsk, Sea of Japan and North Pacific Ocean, and provides more than
2/3 of Russian fish catches (FishNews, 2014, 2015, 2016, 2017) and a large proportion of
catches by Canada, China, Japan, Korea and the USA (FAO, 2010, 2012, 2014; The state, 2002,

2012, 2014, 2016).

Monitoring of marine communities has been carried out in this region for many years by

the Pacific Branch of Russian Federal Research Institute of Fisheries and Oceanography
(TINRO) (Volvenko, 2016). Large amounts of data on nekton, benthos and macroplankton were
collected from trawl catches, and referred to "trawl macrofauna". Under this term we consider
animals with body size from one centimetre to several meters, weighing from several grams to
hundreds of kilograms, and caught by bottom and midwater trawls with fine-mesh liner in the
cod end.

Recently we published (Volvenko et al., 2018) a species checklist of fishes, cyclostomes

and invertebrates recorded during TINRO trawl surveys in the North Pacific and adjacent Arctic
regions (Chukchi Sea) over a period of 38 years. For each species, information was presented on
the type of trawl (benthic and/or midwater) and geographic occurrence.

The main objective of the present study was to analyse the importance of trawl

macrofauna to fisheries. We extended our published checklist (Volvenko et al., 2018) with
commercial and fishery relevant data. Each species entry in the new version of the checklist
(Volvenko et al., 2019) provides the following additional information: Russian and English
common names, real or potential commercial value, potential product yield (percentage of the
raw weight) and minimum wholesale prices in the USD ($) per ton. It is expected that this
information will be useful not only for scientists, but also for fishers, experts in aquaculture,



businessmen, economists, managers and stakeholders in areas of resource management, fishing,
food industry, environmental protection, geopolitics, etc. To our knowledge, this is the first
suchlike study in the North Pacific and in the World Ocean as well.

90         The analysis of the checklist includes a comparison of basins (seas and ocean) and zones

(pelagic and seabed) by proportion of commercial species, and ratio of species with different
product yield values and prices. In conclusion we consider practical application of the checklist.
**Materials and Methods**

94         Material was obtained from databases (Volvenko and Kulik, 2011; Volvenko, 2014), as

well as from the recent trawl surveys conducted by TINRO through 2014. The sampling area
covers nearly 25 million km$^2$ (Table 1). Specimens were collected at 36640 bottom trawl stations
at depths from 5 to 2000 m, and at 32263 midwater trawl stations, mostly at depths from the sea
surface (0 m) to 1000 m, although some mesopelagic hauls reached 2200 m. Both types of trawls
(bottom and midwater) were equipped with a 10-12-mm fine-mesh liner in the cod end. Almost
one billion individuals of various macrofauna species have been recorded in the trawl catches.

101        Published and Internet data were used to further extend the accuracy of information on

the presence (+) or absence (–) of a species in trawl catches in each specific basin by adding
reliable species records not listed in the TINRO databases. In these cases, a species that was
known to occur in a basin but was absent from our samples, was marked with an asterisk (*).
Therefore, in the final checklist, only species never vouchered from a particular basin, based
both on our data and published data, were marked as absent (–). More details on the checklist in
Volvenko et al. (2018).

108        To verify information on geographical distribution, taxonomic status, and accepted

scientific names of species, we used 63 publications (Sasaki, 1929; Kondakov, 1941;
Akimushkin, 1963; Melville and China, 1969; Young, 1972; Zhirmunsky, 1976; Holthuis, 1980;
Nesis, 1982, 1985; Boyle, 1983; Masuda et al., 1984; Roper et al., 1984; Okutani et al., 1987;
Reshetnikov et al., 1989; Williams et al., 1989; Filippova et al., 1997; Shevtsov and Mokrin,



1998; Voss et al., 1998; Borets, 2000; Moiseev and Tokranov, 2000; Norman, 2000;
Mecklenburg et al., 2002; Stepanov et al., 2002; Houart and Sirenko, 2003; National, 2004;
Nelson et al., 2004; Jereb and Roper, 2005, 2010; Kantor and Sysoev, 2005,2006; McLaughlin et
al., 2005; Petryashev, 2005; Katugin and Zuev, 2007; Kosyan and Kantor, 2007, 2009;
Chernova, 2008; Anderson et al., 2009; Organization, 2009; Safran, 2009; Sirenko et al., 2009;
Katugin et al., 2010; Bazhin and Stepanov, 2012; Katugin and Shevtsov, 2012; Sirenko, 2012,
2013; Yavnov, 2012; Baldwin, 2013; Lindberg and Gerd, 2013; Marin, 2013; Shevtsov et al.,
2013; Tuponogov and Snytko, 2013; Danilin, 2014; Jereb et al., 2014; Komatsu, 2014; Mah et
al., 2014; Marin and Kornienko, 2014; Parin et al., 2014; Tuponogov and Kodolov, 2014;
Lebedev, 2015a, b; Lebedev and Tyurin, 2015; Markevich, 2015; Okutani, 2015) and 71 online
resources (Table 2).

At the next stage, the checklist was supplemented with information on the commercial

status of different species obtained from publicly available sources.

Whether a certain species is considered "commercial" in Russia is formally based on

national regulating documents. There are four such orders (The Order, 2009-2011, 2012a, b,
2015). However, not all species listed in those documents are commercially harvested in
practice. That is why we additionally used the recent official statistics on the aquatic biological
resources catch in the Russian Federation (available at http://fish.gov.ru/otraslevaya-
deyatelnost/ekonomika-otrasli/statistika-i-analitika date of latest access October 30, 2018) and
also information contained in the "Fishing" database of the Regional Data Center of the TINRO
(Volvenko, 2015), which particularly includes data for time periods that are not present in the
Center of Fishery Monitoring and Communications (http://cfmc.ru date of latest access October
30, 2018) such as 1980-1994 data from the "RIF" IT system – Russian database on daily fishing
ship reports.

Sometimes it is difficult to ascertain whether a species is harvested commercially in other

countries. For example, only 2% of squids on the world market are identified to species. The rest



are sold under the name "squid", classified as "squid nei" (i.e., not identified to species level)
(Arkhipkin et al., 2015). Nevertheless, we examined existing market price information and sales
information from published and web sources for all the species on our checklist.

Another challenge was distinguishing between non-commercial and potentially

commercial species. For example, several ascidians are known as traditional animals for fishing
and aquaculture (e.g., Lambert et al., 2016). Information on other species is contradictory, from
"all of them are edible too" to "most of them are poisonous". In such cases only species with
confirmed information on their edibility were considered as potentially commercial. However,
we did not consider species to be commercial if they are edible, but too exotic for food industries
in most countries. Some examples of such animals are sipunculids, which are used only in
certain areas of China (https://commons.wikimedia.org/wiki/File:Sipuncula.jpg?uselang=ru date
of latest access October 30, 2018), and also echiurids, which are used exclusively in China and
Korea (https://www.tripadvisor.ru/LocationPhotoDirectLink-g294197-d1842046-i99924342-
Noryangjin_Fish_Market-Seoul.html date of the latest access October 30, 2018). These animals
are also used for making expensive biochemical drugs (http://www.novoprolabs.com/p/urechis-
exitatory-peptide-uep-c-311322.html date of the latest access October 30, 2018); however they
are not subject to large-scale target fishing. Although such species and similar aquatic biological
resources occur widely, the commercial market for them is very small, and the limited demand
that exists is supplied by local catches of native people. We also considered commercial status of
traditional species for recreational and sports fishing, the prices for which are not publically
available, but which are commonly mentioned in mass media.

Technological norms and standards for waste and losses during processing of seafood

were taken from two sources (Basin-scale, 2013, 2014); prices for end products are based on two
electronic periodicals (Far Eastern, 2014-2015; Russian, 2014-2015) and 13 websites (date of the
latest access October 30, 2018):
1.  http://fishretail.ru/monitorings,



2. http://vladivostok.pulscen.ru/price/4005-ryba-moreprodukty,
3. http://www.agroserver.ru/ryba-moreprodukty,
4. http://www.fish.krab.ru,
5. http://www.fishnet.ru,
6. http://www.fishnewseu.com/prices.html,
7. http://www.fishnotice.com,
8. http://www.grimsbyfishmarket.co.uk/fishprices/index.php/prices,
9. http://www.newfultonfishmarket.com/wholesale_price_reports.html,
10. http://www.pulscen.ru,
11. http://www.ru.all.biz/ryba-morskaya-vseh-vidov-bgc143,
12. http://www.rybinfo.ru,
13. http://www.st.nmfs.noaa.gov/commercial-fisheries/market-news.
We used the following algorithm to get minimum wholesale prices for species from the
checklist:
a) First we looked at the price on the Russian market (because it is usually smaller than in other
countries); if the price was not found, we looked at prices of Russian products in Japan and
China (where they are lower than prices of American or European products); if the price was
not found again, then we looked at any price regardless of supplier and market.
b) When there were several market offers, we chose the lowest price.
c) Prices in currency other than U.S. $ (Russian Roubles, Japanese Yen, Chinese Yuan, Euro,
Norwegian Kroner, the UK Pounds, etc.) were recalculated to $ using the exchange rate at the
time of the price publication.
d) When there were prices in different years, months or weeks, we calculated the average of
minimum prices obtained at the previous steps.
e) If the price was not found, we used the price of a similar (analogous) species.
f) If a species has no commercial value *per se* but is suitable at least for being processed into fish
meal, it received the product yield and the price of fish meal.
The resultant rating is more important for fishers than for consumers of fish products:
prices in the list are obviously much lower than retail prices and significantly lower than prices
in restaurants.
**The checklist**
The compiled checklist (Volvenko et al., 2019) is presented in the Supplementary Table.
It includes 1541 rows (corresponding to our minimum estimate for the trawl macrofauna species
richness in the study area) and 14 columns.
The first column shows the scientific name of a species (genus, family) in Latin. Names
are given in alphabetical order. They are not arranged by taxa, which was done in order to
simplify the use of the table by non-experts in taxonomy and even in biology. For example, a
business person or a clerk can find scientific names of interest in (e.g., in the Internet or a
publication) and get information on that species without knowing the details of taxonomy.
In the second and third columns, there are English and Russian common names,
respectively. Species and genus names are given in singular, family names in plural. Russian
names are given for all species in the checklist. Japanese and Chinese names are also known for
all these species, though they are not given for the sake of space. However, English common
names were not found for 167 species, 20 genera and 6 families in the checklist. That is why
there are 193 gaps in the second column.
At the same time, some species may have several common names, even in English. We
listed all the names we could find, and arranged them by frequency of their usage (more
commonly used names are given first). In the second and third columns, names that differ are
separated by commas, e.g., common names of *Argyrosomus japonicus* are "Japanese meagre,
mulloway". Names, which share a common word, are given in parentheses: e.g., for, for
*Argyropelecus sladeni* the common names are shown as "Lowcrest (Sladen's) hatchetfish",
which corresponds to "lowcrest hatchetfish or Sladen's hatchetfish"; therefore, a word outside



parentheses is not repeated for the sake of space. The species *Auxis rochei* has common names
"Bullet tuna (mackerel), bonito", which indicates that it has three names: bullet tuna, bullet
mackerel and bonito. Therefore, parentheses indicate "or", whereas a comma outside parentheses
corresponds to "and". Hence, the names above should read: 1) Japanese meagre and mulloway;
2) Lowcrest or Sladen's hatchetfish; 3) Bullet tuna or mackerel and bonito. The most difficult
case of entry in the second column is when commas inside parentheses indicate "or", e.g.,
"Alaskan (Alaskan bay, Alaskan sand, Northern crangon, salt-and-pepper) shrimp" corresponds
to Alaskan or Alaskan bay or Alaskan sand or Northern crangon or salt-and-pepper shrimp,
meaning five names each consisting of two-to four words: Alaskan shrimp, Alaskan bay shrimp,
Alaskan sand shrimp, Northern crangon shrimp, and salt-and-pepper shrimp.

The fourth column "Taxon" is a numeric code, corresponding to one of 20 aggregate

higher taxonomic groups:
– Fishes;
– Cyclostomes (lampreys and hagfishes);
– Ascidians and pelagic tunicates (salps and appendicularians);
– Crabs (Brachyura) and craboids (lithodids from Anomura);
– Shrimps and crangonids;
– Other crustaceans (hermit-crabs, burrowing mantis shrimps, squat lobsters, isopods,
amphipods, and cirripeds);
– Cephalopods (paper nautiluses, octopuses, squids, and cuttlefishes);
– Gastropods including pelagic ones (heteropods, pteropods, and nudibranchs);
– Bivalves;
– Other molluscs: polyplacophorans (chitons) and solenogasters;
– Sea urchins;
– Sea cucumbers;
– Other echinoderms (brittle stars, starfishes, and sea lilies);



– Coelenterates (jelly-fishes, polyps, corals, sea fans, and anemones);
– Comb jellies;
– Bryozoans;
– Sponges;
– Pycnogonids (pantopods or sea spiders);
– Brachiopods;
– Other invertebrates – this is an aggregate group, which contains the so-called "worms":
annelid polychaetes, flat worms, nemerteans, sipunculans, priapulans, and pogonophorans; they
are rarely found in trawl catches and lack commercial value.

In the fifth column "Mid" and the sixth column "Bot", species occurrence in midwater

and bottom trawl catches is shown, respectively, where "+" corresponds to presence, and "–" to
absence.

Columns from seven to eleven indicate species occurrence in basins, where "C"

corresponds to the Chukchi Sea, "B" – Bering Sea, "O" – Sea of Okhotsk, "J" – Sea of Japan and
"P" – Pacific Ocean. Species presence is indicated by "+", absence by "–", and "*" means
absence from catches but presence according to the published data.

The 12th column "Use" shows commercial use of a species according to the following

five categories:
4 – harvested in Russia based on official statistical reports in 2010-2015;
3 – formerly harvested by Russian fishers or harvested in neighbour countries;
2 – present in official Russian list of commercial species but not harvested in Russia;
1 – not on the list, but potential commercial species;
0 – cannot be used (even for producing of fish meal or fish oil) at the present level of technology.

The 13th column shows potential product yield (the proportion of raw weight). Non-

commercial species are indicated as "0".

In the 14th column minimum retail price is shown (in $ per ton). Non-commercial species





are indicated as "0".
The last three columns are explained in Material and Methods.
**Proportion of commercial species in the fauna**
Parin et al. (2014) indicated that the number of commercial fish species in the fauna of
Russia varied from 250 to 700 according to various published sources. The authors themselves
included only 145 species into this category, having noted that only about 50 species (<4% of the
total fish fauna list) can be considered as true targets for large-scale fishery (Parin et al., 2014, p.

559).

According to our data (Table 3), based on official reports from fishing companies, only in
the North Pacific and East Arctic, Russian fishers actually harvest 329 fish species that are
included in our list (Supplementary Table). When considering prospective commercial species
that occur in the study area, including species fished in other countries, the number of
commercial fish species will increase up to 860, which accounts for about 50% of the total fish
fauna list. Actual and potential number of commercial invertebrate species are almost two times
lower: 173 and 374 species respectively.
The analysis of distribution of species from different higher taxa across five fishing status
categories (classification is presented in the Checklist section) (Table 3, Fig. 2) showed that, at
the modern level of development of science and technology, only 20% of trawl macrofauna
species, including 9% fish species and 36% invertebrates, have no practical commercial value at
all (category 0). Invertebrates in this category were dominated by echinoderms (20%) and
molluscs (12%). Crustaceans, coelenterates and benthic invertebrates classified as "other" group
accounted for 8%, the remaining comprised ≤5%. Altogether, they formed the vast majority
(71%) of non-commercial species, and the remaining 29% of species in this category were fish.
The opposite pattern was observed among commercially harvested species (categories 1-
4): fishes accounted for 64-78% (depending on the category) of all species, and invertebrates 20-
36%. The latter group was dominated by molluscs and crustaceans.



The number of species of fish, invertebrates and total trawl macrofauna consistently
decreased from the 4th to the 2nd category (Fig. 2), suggesting that most species (502) were
harvested by fishers in Russia and other countries. Less number of species (185) were harvested
by non-Russian fishers (because these species are rare in the Russian EEZ and/or are not
traditional targets for the Russian fishing industry). Lastly, very few (39) species were formally
listed as commercial in fishing regulating documents.
Of particular interest is the large number of species (512), which have potential
commercial importance. Among them 71% are fishes, 14% shellfish, 11% crustaceans, and 2%
sea cucumbers and jellyfish, many of which are suitable for human consumption, feeding of
animals, production of fish meal, fish oil, and a wide variety of other uses. However, there is no
specialized fishery for those species, and when they are caught as by-catch they are discarded
because they are rare or poorly-known for fishers. Some species are abundant, but the
commercial value is low or they require economically impractical processing. These 33% of
species in the checklist constitute untapped potential for commercial fishing in the study area.
The appearance of a species in a specific commercial category undoubtedly depends on
its market value. However, these relationships (Fig. 3) are not straightforward. The price is
necessary but not sufficient condition for placement of a species into a certain category of use:
inexpensive and even very cheap species are present in all categories.
All species, the wholesale prices for which are >$20 thousand per ton, are used by
Russian fisheries (category 4). These include invertebrates (in descending order of price):
*Apostichopus japonicus, Eriocheir japonica, Pandalus hypsinotus, Paralithodes camtschaticus*,
and *Lithodes aequispinus*.
Species worth ≤ $20 thousand per ton appear in both the 4th and 3rd categories. For
example, the most expensive among them ($20 thousand per ton) - *Sclerocrangon boreas, S.*
*derjugini, S. salebrosa* and *Mesocrangon intermedia*, are harvested by Russian fishers, whereas
*Lithodes couesi* is not. The latter species is too rare and not abundant in traditional fishing areas.



The same is true for most other fishing targets from the third category. In particular, species with
wholesale values of $10-15 thousand per ton: *Anguilla* sp., *Thunnus orientalis, Parvamussium*
*alaskense* and *Chionoecetes tanneri*, certainly would have been harvested by Russian fishers if
they were physically (geographically) available to them.

It is noteworthy that in the 2nd category (formally commercial species), only species of

relatively low value appeared: fishes worth < $3000 per ton (mainly anglefishes, goosefishes,
dreamers, wolffishes, sticklebacks, and one of the most abundant mesopelagic fish species of the
surveyed area northern smoothtongue *Leuroglossus schmidti*) and invertebrates valued less than
$1500 per ton (squids of the genus *Gonatus*, bivalves and jellyfish).

Species with the same commercial value appear among those that are actually harvested

(categories 4 and 3). Among potentially commercial species (category 1), species with the same
or higher commercial value were present, e.g., ascidians, fishes and shellfish. However, we were
not able to find any information on fishery or sale prices for them in the literature (their potential
prices were determined by comparison with similar commercial species).

The first category included edible ascidians of the genus *Boltenia* potential wholesale

prices for which (by comparing with *Styela clava*, *Halocynthia aurantium,* and *H. roretzi*) can
exceed $3000 per ton; small squid valued at < $1500 per ton; gastropods, bivalves, crabs,
shrimps, holothurians and jellyfishes worth < $1000 per ton. This category also includes many
fish species, including chimaeras, sharks, mackerels, and pomfrets, for a total of 28 species
worth ≥ $1000 per ton, and a large number of fish species (334) worth < $1000 per ton, in
particular poachers, eelpouts and lantern fishes. Potential commercial stocks of some of these
species are very large. For example, biomass values for small mesopelagic fishes and squid were
estimated at hundreds of millions of tons (Gjosaeter and Kawaguchi 1980, Karedin, 1998,
Beamish et al., 1999, Irigoien et al., 2014, Shuntov, 2016).

345       Marine basins and different zones were analysed separately (Table 4) and compared with

each other (Fig. 4).





347  Among non-commercial species (category 0), fewer species occurred in the pelagic zone

348 (12 to 15%) than on the bottom (22 to 26% in different areas). In other categories, such

349 differences are not as large: 41-65% of actually harvested species (categories 3 and 4) occurred

350 in the pelagic zone, and 44-51% in the benthic zone; 2-6% and 2-3% of formally commercial

351 species (category 2) occur in the pelagic zone and on the bottom, respectively; and for potential

352 commercial species (category 1), these figures are 19-44% and 24-28%, respectively.

353  Distribution patterns for commercially important species from different categories in

354 different marine basins and biotopes were related to their richness values (Volvenko et al., 2018).

355 The number of both real and potential important commercial species was higher on the seafloor

356 than in the pelagic zone. Among the large marine areas, the highest species richness indices were

357 observed in the Pacific Ocean, followed by Sea of Okhotsk, then Bering Sea and Sea of Japan

358 (with minor differences among these two seas), and with the lowest values in the Chukchi Sea.

359  Species richness in the Sea of Japan may be underestimated due to the relatively small

360 sample size (Table 1), and therefore the number of commercial species that occur in that basin

361 may in fact be larger, taking into account that the number of (actual, formal and potential)

362 commercial species increases from northern to southern basins along with species richness

363 values (Volvenko et al., 2018).

364 **Species ranking by potential product yield**

365  For this analysis, commercial species were subdivided into four groups by potential

366 product output (proportion of raw weight): <0.3; 0.3-0.6; 0.6-0.9 and 0.9-1.

367  In accordance with the actual data (Supplementary Table), the first group included 3% of

368 the species which indicated a potential yield value of 0.1-0.25, the second group included 8% of

369 the species with a potential yield of 0.3-0.6, the third 12% of the species with a yield 0.7-0.9, and

370 the fourth 77% with yield 0.9-1.0 (Table 3, Fig. 5).

371  There were no fishes in the 1[st] group and only seven fish species in the 2[nd] group. These

372 included Pacific cod *Gadus macrocephalus,* escolar *Lepidocybium flavobrunneum*, and some



sharks. The 3$^{rd}$ group, comprising 48 fish species, consists of flatfishes and Pacific salmon. All
other commercial fishes (805 species) belong to the 4th group with the maximum output of raw
products. All (100%) species of cyclostomes also belong to this group.
The number of invertebrate species also increased from the 1$^{st}$ group to the 4$^{th}$, but not so
sharply as fishes (see the right graph in Fig. 5). The 1$^{st}$ group included only gastropods and
bivalves with thick massive shells. In the 2$^{nd}$ group crab species appeared those with only limbs
on sale. The 3$^{rd}$ group was dominated by cephalopods (it also included most of bivalves and
holothurians), and in the 4$^{th}$ group there are mainly crustaceans.
All species of ascidians, pandalid and crangonid shrimps, other crustaceans, other
molluscs, sea urchins and jellyfishes were also included into the 4$^{th}$ group. This group also
includes 81% of crab species. All these are invertebrates with a maximum product yield. The
minimum yield of production is characteristic for shell molluscs: 77% of gastropods belong to
the 2$^{nd}$ group and 23% to the 1$^{st}$ group; 63% of bivalves are in the 3$^{rd}$ group, 16% in the 2$^{nd}$
group, 18% in the 1$^{st}$ group, and only 4% in the 4$^{th}$ group.
We further considered basins and zones separately (Table 4) and compared them (Fig. 6).
There were significant differences between pelagic and benthic zones in all "technological"
groups. In the pelagic zone in different basins, species with a minimum production output (group
1) account for 0 to 0.4% of all species, whereas on the seafloor from 4 to 8%. In the 2$^{nd}$ group
there are 1-2% of pelagic species and 9-14% of benthic species; in the 3$^{rd}$ group 12-14% of
pelagic species and 14-19% of benthic species. Therefore, the proportion of species with the
maximum product yield of production from raw material (group 4) is much higher in the pelagic
zone (83-87%) than on the seafloor (60-73%). This is explained by differences in fauna of the
water column and the seafloor: the most high-tech species are pelagic nektonic fish, shrimp and
cephalopods, whereas on the seafloor, many invertebrates, such as shelled molluscs, are
characterized by comparatively low production yield.
The following patterns of species distribution into different "technological" groups in all



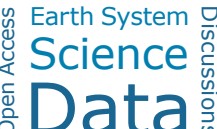

399 studied basins were revealed (Figs 5, 6, Table 4):

400 1) The higher the product yield, the higher the number of species in a group;

401 2) The majority of species with low product yield occur on the seafloor;

402 3) The number of species in each "technological" group generally corresponds to species

403 richness in different marine basins, and in most cases, species richness increases from northern

404 to southern basins, with the exception of the Sea of Japan that has been mentioned already in our

405 previous publication (Volvenko et al., 2018).

406 **Species distribution by price range**

407 To analyse the distribution of species by price, we ranked them into six price categories:

408 $0, $1000, $2000, $5000, $10000 and $20000 per ton. In accordance with the data

409 (Supplementary Table), all representatives of trawl macrofauna fell into seven uneven, in terms

410 of species richness, price categories (Table 3, Fig. 7):

411 1) zero price, i.e. non-commercial ($0 per ton) – 303 (20%) species,

412 2) very cheap ($500-900 per ton) – 597 (39%) species,

413 3) cheap (($1000-2000 per ton) – 197 (13%) species,

414 4) on the average inexpensive (2.1-5 thousand $ per ton) – 279 (18%) species,

415 5) on the average price (5.4-9.2 thousand $ per ton) – 114 (7%) species,

416 6) expensive (10-15 thousand $ per ton) – 41 (almost 3%) species,

417 7) very expensive (20-30 thousand $ per ton) – 10 (less than 1%) species.

418 Therefore, it appeared that more than half of the number of species (58%), which were

419 captured in a trawl, fell into non-commercial and very cheap category, the price for which is less

420 than 0.9 thousand $ per ton.

421 The 1st category is dominated by invertebrates, whereas the 2nd by fishes. Fishes also

422 dominate among "cheap" and "on the average inexpensive" commercial target species, and

423 invertebrates among "on the average pricy", "expensive" and "very expensive". All cyclostomes

424 belong to 4th and 5th price categories (Fig. 7, right graph).



Prawns and shrimps (*Pandalus hypsinotus, Mesocrangon intermedia* and representatives

of the genus *Sclerocrangon*), crabs (mokuzu and king crabs) and holothurian (actually one
species *Apostichopus japonicus*) comprised, respectively, 50%, 40% and 10% in the category of
"very expensive" commercial species, the lowest in species richness

Shrimps and crangonids account for 46%, bivalves (scallops and blood clam *Anadara*

*broughtonii*) 24%, crabs (primarily the genus *Chionoecetes*) 15% and fishes 15% of "expensive"
commercial species.

The category "on the average pricy" consisted mainly of gastropods that belong to the

family Buccinidae (71%), with fishes (11%), sea urchins (7%) and crabs (4%) being less
important. Cyclostomata (hagfish), Ascidia and Cephalopoda (octopuses) each comprise 2%, and
ocean and mantis shrimps only 1%. In general this price category was dominated by
invertebrates. The ratio of invertebrates is higher only among the cheapest and non-commercial
species.

Fishes accounted for 89%, cephalopods 4%, bivalves 3%, sea cucumbers 2%,

cyclostomes (lampreys) 1%, and ascidians 1% of the category "on the average inexpensive".

Fishes accounted for 69.5%, squid 22%, bivalves 6% and shrimps 2.5% of the category

"cheap".

Fishes accounted for 77%, and different higher taxa of invertebrates 0.2-6.7% of the

category "very cheap". Only tunicates, cephalopods and sea urchins were absent from this group,
because they are more expensive. As it was mentioned above, that price category had the highest
species richness, and contained 56% of all commercial crustaceans, 22% gastropods, 45%
bivalves, 100% other molluscs, 63% holothurians and 100% jellyfish.

The revealed pattern in a number of species and groups in the price categories generally

remains in different basins and zones (Table 4, Fig. 8).

Notable differences between the pelagic zone and bottom exist with respect to percentage

of "on the average pricy" species: they account for 9-15% of species on the bottom in different





basins, and 2-3% in the pelagic layer. The percentages of other price categories were pretty
similar in the pelagic zone and on the bottom, e.g., "very cheap" species accounted for 39-54%
and 46-53%, "cheap" 11-22% and 11-14%, "on the average inexpensive" 22-33% and 15-25%,
"expensive" 2-8% and 4-8%, and "very expensive" 0.7-3.0 and 1.2-1.5%% in the pelagic zone
and on the bottom, respectively.

The number of "expensive" and "very expensive" species (with a price range 10-30

thousand $ per ton), captured in bottom trawl hauls, gradually increased from north to south, and
in the Okhotsk Sea pelagic trawl hauls, that number was higher than in the ocean (Fig. 8).
Distribution of species that belong to other price categories on the bottom and in the pelagic zone
echoes distribution of the total species richness in different basins (see Volvenko et al., 2018).
**3D distribution of species by fishing, technological and price groups**

462         Distribution of trawl macrofauna by commercial categories, production output and prices

was further analysed in a 3-dimensional space (Fig. 9).

464         Theoretically, the higher the product output and the price of a product from a species on

the market the higher the commercial value of that species. Species with the highest commercial
values are located in the far upper corner of the cube of coordinates (Fig. 9). Somewhat more
than ten species, most of which are invertebrates, are in that corner. Total catch of those species
was, is and will be relatively low. Species with reverse properties that have low output of cheap
production are located in the opposite part of the system of coordinates, in the near lower corner.
There, close to non-commercial species, are small gastropods, which we consider as potentially
commercial and, probably, the most unattractive for fishery in that category. However, locals
collect them, cook and sell on street markets in the Southeast Asia (similar to fried insects), and
of course, these small animals are not harvested in large quantities due to relatively low market
requirement. Total biomass of fishes and invertebrates that have very low and zero value is many
times higher than the existing total catch of commercial aquatic biological resources, and
potential interest in that group of potentially commercial biological resources may appear in



future with the Earth population growth. In that case, more than 500 points, which aggregate in
the left part of the cube, will shift to the right, which means that potentially valuable species will
become commercial. Some of 303 points, located at the root of coordinates (0;0;0), which
correspond to non-commercial and out-of-use species of trawl macrofauna, may also change
their position in future with the development of science and technology.

482         At present, more than a half (687) of commercial species in the examined area are

harvested (Fig. 9). Most of the species here are technologically profitable (production yield
exceeds 0.9); however, they are inexpensive (price is less than 10 thousand $ per ton). In
particular, walleye pollock *Theragra chalcogramma*, which is the leader in terms of the catch
amount and inexpensive (less than 2 thousand $ per ton) is located in the lower quarter of the far
corner on the graph together with other relatively cheap fish, such as Pacific herring *Clupea*
*pallasii,* pink salmon *Oncorhynchus gorbuscha,* Japanese sardine *Sardinops melanosticta,*
Saffron cod *Eleginus gracilis,* greenlings *Pleurogrammus* spp., Pacific saury *Cololabis saira,*
capelin *Mallotus villosus*, and flounders, which comprise the basis for fishery harvest in Russia
and many other countries (Fig. 9).

492         It is clear that, besides the production output and price, there are other factors, which may

influence commercial status of a species. These are primarily 1) commercial stock abundance, 2)
stock availability for fishery, and 3) market demand, i.e. feasible sales rate. Therefore, the graph
would be more informative in case the categorical scale for species commercial importance
along the X-axis is replaced by continuous scale showing amount of their annual catch.
Unfortunately, so far we do not have such kind of data for most of the 1541 species listed in
Supplementary Table.

499         In this review, we did not consider the issue of commercial use of different parts and

organs of marine organisms, e.g., production output for liver and eggs of cod, pollock, herring,
salmonids, flying fish, etc., the price for which significantly exceeds the price for the fish itself
in some countries. This is the scope for future research.



503       Finally, it should be noted that most numerous species, which dominate in the surveyed

region, are usually r-strategists (as defined by MacArthur and Wilson, 2001), i.e. they are
characterized by relatively low competitiveness, high breeding performance and frequency of
reproduction, absence of care for their offspring, small size, fast development and short life
cycle, strong dependence of fertility and mortality on the influence of external factors. Therefore,
they are characterized by perennial cyclical fluctuations in abundance – the so-called "life
waves" with periods from several years to several decades. Such fluctuations were reported for
highly abundant commercial fish: anchovies, herrings, pollock, salmon, mackerel, scad, sardines,
etc. (see e.g.: Davydov, 1986, Shuntov, 1986, 2000, 2016, Klyashtorin and Lyubushin, 2005).
Therefore, the sustainable fishery in the region can be achieved only by expansion of the
assortment of commercially used bioresources. The supply of bioresources in the far Eastern seas
and North Pacific provides such opportunity.
**Data availability**
Volvenko et al., , 2019
**Conclusions**

The analysis of the trawl macrofauna checklist we continued in the present study (the first

part in Volvenko et al., 2018) yielded several practical outcomes:
1) Almost 20% of species in trawl catches (the percentage is higher at the seafloor than in the

pelagic zone) were non-commercial species, and >50% were cheap or very cheap with price

ranging from 0.5 to 2 $/kg. Among the latter, fish species were the most intensively harvested

in the region.

2) Only 3.3% of all species belonged to expensive and very expensive (10-30 $/kg) commercial

525       species. These categories were dominated by invertebrates: Japanese sea cucumber, shrimps,

526       crabs and scallops. The number of such species increased from northern to southern basins.

3) Of all examined species of the trawl macrofauna, 33% can be considered as unexploited

528       reserve for the fishing industry. These are mainly small fish, squids, shrimps and benthic



invertebrates, with their total biomass many times exceeding that of currently fished biological
resources.
4) Most of potentially commercial species were technologically highly profitable (with the
product output exceeding 0.9 of the raw weight). The percentage of such species was much
higher in the pelagic zone (dominated by "profitable" fish, cephalopods and shrimps) than on
the seafloor (with many invertebrates that have low product yield, in particular, the shelled
molluscs).
5) Product yield and price are necessary but not sufficient conditions for including species into a
certain category of commercial use, and do not necessarily reflect catch amount. They also
depend on commercial stock abundance, its accessibility for fishery and market requirement,
i.e. potential sales rates.
6) It is known that the most abundant commercial species in the Far Eastern region are subject to
significant natural fluctuations in the abundance, therefore the sustainable fishery in the region
can only be secured by expansion of the assortment of commercial bioresources. The supply of
bioresources in the far Eastern seas and North Pacific provides such opportunity.
In the future, more valuable information can be obtained from the checklist we presented
using other methods of data processing and/or additional data (such as abundance, occurrence
and catches). Comparisons with similar checklists from other areas or with checklists from the
same area obtained using different techniques also may be of interest.
We hope that our checklist of fauna will be helpful to ichthyologists, hydrobiologists,
ecologists, biogeographers, conservation biologists, economists and fishery managers, as well as
to teachers and students of respective specialties. Potential fields of practical use of the checklist
may include: management of living marine resources, aquaculture development and nature
conservation. In particular, it can be used to assess economic value of biological resources,
which was done, e.g., in the Sea of Okhotsk (Lukyanova et al., 2016), or damages to marine
ecosystems resulting from anthropogenic impact, including pollution, hydro-technical



constructions, oil and gas extraction, tanker or nuclear reactors accidents, etc. In the simplest
case, in order to estimate such damage in terms of cost, total destructed biomass should be
multiplied by possible product yield and prices for respective species shown in the present study.
For more comprehensive assessment, the same procedure should be conducted taking into
account the potential offspring of these animals over a certain period of time, and resulting
amounts should be summed up.
**Author contribution**
IVV, planning and coordination of work, database creation, data analysis, preparation of all
tables and figures, writing a manuscript text; AMO, checking and editing the list of fish and
cyclostomes, adding and editing the text of the manuscript; AVG, checking and editing the list of
all invertebrates except cephalopods, adding and editing the text of the manuscript; ONK,
checking and editing the list of cephalopods, editing the text of the manuscript; AAO, collection
of data on prices and product yield; GMV, collecting of literature data on invertebrates; OAM,
collecting literature data on fishes.
**Competing interests**
The authors declare that they have no conflict of interest.
**Acknowledgements**

We are grateful to Prof. V.P. Shuntov (TINRO) for valuable critical notes made during

preparation of the manuscript, to Dr. A.V. Sysoev (Zoological Museum, Moscow State
University) for comments on common names of gastropods, and to S. Wildes (Auke Bay Labs,
Juneau, Alaska) for a review of the manuscript. The contribution of A.M. Orlov to this
publication was partially supported by the Russian Fund of Fundamental Research (projects Nos.
16-04-00516 and 16-04-00456).

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



**Table 1** Parameters of samples used to generate the checklist (from Volvenko et al., 2018).

| Basin | Zone | Survey years | Depth range, m | Number of stations | Study area, thousand km² | Total trawling time, hours | Total sampling area, km² | Number of individuals sampled |
|---|---|---|---|---|---|---|---|---|
| Chukchi Sea | pelagic | 2003-2014 | 0-91 | 239 | 298 | 162 | 40 | 1 701 314 |
| | benthic | 1995-2014 | 13-222 | 237 | 286 | 118 | 10 | 631 531 |
| | combined | 1995-2014 | 0-222 | 476 | 298 | 280 | 50 | 2 332 845 |
| Bering Sea | pelagic | 1982-2014 | 0-920 | 4 959 | 1 419 | 5 939 | 1 966 | 68 718 728 |
| | benthic | 1977-2014 | 6-1400 | 9 235 | 1 028 | 6 608 | 901 | 23 978 418 |
| | combined | 1977-2014 | 0-1400 | 14 194 | 2 126 | 12 547 | 2 867 | 92 697 146 |
| Sea of Okhotsk | pelagic | 1980-2014 | 0-1000(2200) | 11 053 | 1 523 | 10 598 | 3 232 | 98 376 567 |
| | benthic | 1977-2014 | 5-2000 | 10 073 | 1 385 | 7 159 | 819 | 33 190 559 |
| | combined | 1977-2014 | 0-2200 | 21 126 | 1 523 | 17 757 | 4 051 | 131 567 126 |
| Sea of Japan | pelagic | 1981-2013 | 0-720 | 2 621 | 447 | 2 456 | 836 | 34 663 510 |
| | benthic | 1978-2014 | 5-935 | 10 766 | 137 | 6 235 | 591 | 13 593 004 |
| | combined | 1978-2014 | 0-935 | 13 387 | 447 | 8 691 | 1 428 | 48 256 514 |
| Pacific Ocean | pelagic | 1979-2014 | 0-1000(1230) | 13 391 | 17 741 | 19 859 | 7 720 | 538 822 020 |
| | benthic | 1977-2012 | 10-1860 | 6 329 | 1 262 | 8 150 | 1 498 | 34 732 062 |
| | combined | 1977-2014 | 0-1860 | 19 720 | 20 236 | 28 009 | 9 217 | 573 554 082 |
| Total area | pelagic | 1979-2014 | 0-2200 | 32 263 | 21 429 | 39 014 | 13 794 | 742 282 139 |
| | benthic | 1977-2014 | 5-2000 | 36 640 | 4 097 | 28 271 | 3 819 | 106 125 574 |
| | combined | 1977-2014 | 0-2200 | 68 903 | 24 630 | 67 285 | 17 613 | 848 407 713 |

Footnote: Maximum depth (at which only few trawls were taken) is shown in parentheses; study area included all trawl stations and was calculated by contouring areas with stations (see Fig. 1); total sampling area was calculated as the sum of areas covered by trawl hauls; area of a trawl haul was calculated by multiplying trawl horizontal opening by trawling distance.





**Table 2** The list of web-links providing information on taxonomy, species geographic distribution and common names.

| N | URL (Uniform Resource Locator) |
|---|---|
| 1 | http://akully.ru |
| *2 | http://animaldiversity.org |
| 3 | http://aqualib.ru |
| 4 | http://arctos.database.museum/taxonomy.cfm |
| 5 | http://argus.aqualogo.ru |
| 6 | http://bie.ala.org.au |
| 7 | http://bioportal.naturalis.nl |
| 8 | http://bryozone.myspecies.info |
| 9 | http://bvi.rusf.ru |
| 10 | http://bvi.rusf.ru/taksa/alfy/russian.htm |
| 11 | http://calyptraeids.myspecies.info |
| 12 | http://clade.ansp.org/obis/find_mollusk.html |
| 13 | http://collections.nmnh.si.edu/search/iz |
| 14 | http://collections.peabody.yale.edu/search/Search |
| 15 | http://dic.academic.ru |
| 16 | http://eol.org |
| 17 | http://eunis.eea.europa.eu |
| 18 | http://fauna-flora.ru |
| 19 | http://fish.dvo.ru |
| 20 | http://fish.gov.ru/otraslevaya-deyatelnost/ekonomika-otrasli/statistika-i-analitika |
| 21 | http://fishindex.blogspot.sg/ |
| 22 | http://glgolub.narod2.ru |
| 23 | http://ispecies.org |
| 24 | http://marinebio.org |
| 25 | http://nature.legio.in |
| 26 | http://polychaetes.lifewatchgreece.eu |
| 27 | http://ribovodstvo.com/books/item/f00/s00/z0000004/index.shtml |
| 28 | http://shark-references.com |
| 29 | http://sheric.ru |
| *30 | http://slovarbio.ru |
| 31 | http://species-identification.org |
| 32 | http://taxonomicon.taxonomy.nl |
| *33 | http://taxonomy.e-science.ru |
| 34 | http://tolweb.org |
| 35 | http://webapp1.dlib.indiana.edu/virtual_disk_library/index.cgi/4970813/FID2752/html/ecosys/species/lists/inverts.htm |
| 36 | http://www.annelida.net |
| 37 | http://www.apus.ru |
| 38 | http://www.arcodiv.org |
| 39 | http://www.bagniliggia.it/WMSD/WMSDsearch.htm |
| 40 | http://www.biolib.cz/en |
| 41 | http://www.calacademy.org/scientists/projects/catalog-of-fishes |
| 42 | http://www.catalogueoflife.org |
| 43 | http://www.conchology.be |
| 44 | http://www.crabs.ru/ |
| 45 | http://www.fao.org/figis/geoserver/factsheets/species.html |
| 46 | http://www.fegi.ru/primorye/atlas |
| 47 | http://www.fishbase.org |
| 48 | http://www.fishesofaustralia.net.au |
| 49 | http://www.gastropods.com |
| 50 | http://www.gbif.org |
| 51 | http://www.godac.jamstec.go.jp |





| 52 | http://www.inaturalist.org |
| 53 | http://www.itis.gov |
| 54 | http://www.iucnredlist.org |
| *55 | http://www.lifecatalog.ru/cont/animalia.html |
| 56 | http://www.marinespecies.org |
| 57 | http://www.marlin.ac.uk/biotic/ |
| *58 | http://www.molluscsoftasmania.net |
| 59 | http://www.multitran.ru |
| 60 | http://www.octe.ru |
| *61 | http://www.sealifebase.fisheries.ubc.ca |
| 62 | http://www.sealifebase.org |
| 63 | http://www.shellsandsnails.info |
| 64 | http://www.species-identification.org |
| 65 | http://www.squali.com |
| 66 | http://www.ubio.org |
| 67 | http://www.uniprot.org |
| 68 | http://www.zin.ru/zoodiv |
| 69 | http://zooclub.ru |
| 70 | https://en.wikipedia.org |
| 71 | https://ru.wikipedia.org |

Footnote: Web-sites are given in alphabetical order of URLs; date of the latest access to most sites was October 30, 2018, and for those sites whose numbers are marked with asterisks it was January 31, 2018 (now they are no longer available, at least from the territory of the Russian Federation).





**Table 3** Species number of main taxonomic groups in different fisheries, technological and price categories.

| Taxon/Group | Total species | Commercial category | | | | | Product yield (% of raw weight) | | | | Price category (thousand US $ per one ton) | | | | | |
|---|---|---|---|---|---|---|---|---|---|---|---|---|---|---|---|---|
| | | 4 | 3 | 2 | 1 | 0 | 0.1-0.25 | 0.3-0.6 | 0.7-0.9 | 0.92-1 | 0.5-0.9 | 1-2 | 2.1-5 | 5.4-9.2 | 10-15 | 20-30 |
| Fish | 949 | 329 | 144 | 25 | 362 | 89 | 0 | 7 | 48 | 805 | 458 | 137 | 247 | 12 | 6 | 0 |
| Cyclostomes | 4 | 0 | 4 | 0 | 0 | 0 | 0 | 0 | 0 | 4 | 0 | 0 | 2 | 2 | 0 | 0 |
| Tunicates | 21 | 2 | 1 | 0 | 2 | 16 | 0 | 0 | 0 | 5 | 0 | 0 | 3 | 2 | 0 | 0 |
| Crabs and craboids | 36 | 11 | 5 | 0 | 20 | 0 | 0 | 7 | 0 | 29 | 20 | 0 | 1 | 5 | 6 | 4 |
| Shrimps | 70 | 32 | 2 | 0 | 36 | 0 | 0 | 0 | 0 | 70 | 40 | 5 | 0 | 1 | 19 | 5 |
| Other crustaceans | 25 | 0 | 1 | 0 | 1 | 23 | 0 | 0 | 0 | 2 | 1 | 0 | 0 | 1 | 0 | 0 |
| Cephalopods | 85 | 5 | 15 | 7 | 30 | 28 | 0 | 0 | 54 | 3 | 0 | 43 | 12 | 2 | 0 | 0 |
| Gastropods | 109 | 80 | 1 | 0 | 23 | 5 | 24 | 80 | 0 | 0 | 23 | 0 | 0 | 81 | 0 | 0 |
| Bivalves | 57 | 33 | 5 | 3 | 15 | 1 | 10 | 9 | 35 | 2 | 25 | 12 | 9 | 0 | 10 | 0 |
| Other molluscs | 5 | 0 | 0 | 0 | 2 | 3 | 0 | 0 | 0 | 2 | 2 | 0 | 0 | 0 | 0 | 0 |
| Sea urchins | 8 | 7 | 1 | 0 | 0 | 0 | 0 | 0 | 0 | 8 | 0 | 0 | 0 | 8 | 0 | 0 |
| Sea cucumbers | 16 | 2 | 4 | 0 | 10 | 0 | 0 | 0 | 10 | 6 | 10 | 0 | 5 | 0 | 0 | 1 |
| Other echinoderms | 61 | 0 | 0 | 0 | 0 | 61 | 0 | 0 | 0 | 0 | 0 | 0 | 0 | 0 | 0 | 0 |
| Coelenterates | 42 | 1 | 2 | 4 | 11 | 24 | 0 | 0 | 0 | 18 | 18 | 0 | 0 | 0 | 0 | 0 |
| Comb-jellies | 3 | 0 | 0 | 0 | 0 | 3 | 0 | 0 | 0 | 0 | 0 | 0 | 0 | 0 | 0 | 0 |
| Bryozoans | 8 | 0 | 0 | 0 | 0 | 8 | 0 | 0 | 0 | 0 | 0 | 0 | 0 | 0 | 0 | 0 |
| Sponges | 15 | 0 | 0 | 0 | 0 | 15 | 0 | 0 | 0 | 0 | 0 | 0 | 0 | 0 | 0 | 0 |
| Pycnogonids | 1 | 0 | 0 | 0 | 0 | 1 | 0 | 0 | 0 | 0 | 0 | 0 | 0 | 0 | 0 | 0 |
| Brachiopods | 1 | 0 | 0 | 0 | 0 | 1 | 0 | 0 | 0 | 0 | 0 | 0 | 0 | 0 | 0 | 0 |
| Other benthic invertebrates | 25 | 0 | 0 | 0 | 0 | 25 | 0 | 0 | 0 | 0 | 0 | 0 | 0 | 0 | 0 | 0 |
| All invertebrates | 588 | 173 | 37 | 14 | 150 | 214 | 34 | 96 | 99 | 145 | 139 | 60 | 30 | 100 | 35 | 10 |
| Total macrofauna | 1541 | 502 | 185 | 39 | 512 | 303 | 34 | 103 | 147 | 954 | 597 | 197 | 279 | 114 | 41 | 10 |

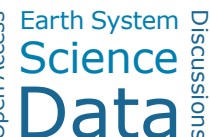

**Table 4** Number of species from different basins and zones in fisheries, technological and price categories.

| Basin | Zone | Total species | Commercial category | | | | | Product yield (% of raw weight) | | | | Price category (thousand US $ per one ton) | | | | | |
|---|---|---|---|---|---|---|---|---|---|---|---|---|---|---|---|---|---|
| | | | 4 | 3 | 2 | 1 | 0 | 0.1-0.25 | 0.3-0.6 | 0.7-0.9 | 0.92-1 | 0.5-0.9 | 1-2 | 2.1-5 | 5.4-9.2 | 10-15 | 20-30 |
| Chukchi Sea | pelagic | 113 | 62 | 5 | 6 | 27 | 13 | 0 | 2 | 14 | 84 | 53 | 11 | 22 | 3 | 8 | 3 |
| | benthic | 274 | 123 | 9 | 7 | 67 | 68 | 16 | 28 | 39 | 123 | 101 | 23 | 32 | 31 | 16 | 3 |
| | combined | 279 | 123 | 9 | 8 | 69 | 70 | 16 | 28 | 39 | 126 | 104 | 23 | 32 | 31 | 16 | 3 |
| Bering Sea | pelagic | 306 | 130 | 14 | 19 | 100 | 43 | 1 | 3 | 38 | 221 | 143 | 39 | 61 | 4 | 12 | 4 |
| | benthic | 679 | 268 | 31 | 22 | 183 | 175 | 23 | 53 | 79 | 349 | 265 | 64 | 85 | 55 | 28 | 7 |
| | combined | 698 | 269 | 31 | 23 | 197 | 178 | 23 | 53 | 80 | 364 | 280 | 65 | 85 | 55 | 28 | 7 |
| Sea of Okhotsk | pelagic | 375 | 155 | 29 | 21 | 112 | 58 | 1 | 4 | 45 | 267 | 159 | 46 | 87 | 7 | 14 | 4 |
| | benthic | 824 | 353 | 42 | 21 | 213 | 195 | 23 | 86 | 95 | 425 | 302 | 75 | 125 | 88 | 31 | 8 |
| | combined | 853 | 355 | 49 | 24 | 224 | 201 | 23 | 86 | 96 | 447 | 315 | 76 | 131 | 90 | 32 | 8 |
| Sea of Japan | pelagic | 265 | 128 | 45 | 6 | 51 | 35 | 1 | 4 | 33 | 192 | 90 | 44 | 75 | 7 | 10 | 4 |
| | benthic | 644 | 278 | 48 | 10 | 156 | 152 | 33 | 55 | 83 | 321 | 232 | 56 | 108 | 56 | 33 | 7 |
| | combined | 678 | 280 | 67 | 10 | 160 | 161 | 33 | 56 | 84 | 344 | 237 | 65 | 117 | 58 | 33 | 7 |
| Pacific Ocean | pelagic | 701 | 172 | 113 | 23 | 306 | 87 | 1 | 6 | 72 | 535 | 308 | 135 | 142 | 13 | 12 | 4 |
| | benthic | 1057 | 396 | 95 | 30 | 301 | 235 | 32 | 71 | 117 | 602 | 382 | 119 | 203 | 76 | 32 | 10 |
| | combined | 1342 | 406 | 166 | 35 | 469 | 266 | 32 | 73 | 142 | 829 | 522 | 189 | 241 | 81 | 33 | 10 |
| Total area | pelagic | 751 | 195 | 120 | 27 | 312 | 97 | 1 | 6 | 73 | 574 | 326 | 138 | 159 | 13 | 14 | 4 |
| | benthic | 1246 | 491 | 113 | 31 | 342 | 269 | 34 | 101 | 122 | 720 | 453 | 127 | 238 | 109 | 40 | 10 |
| | combined | 1541 | 502 | 185 | 39 | 512 | 303 | 34 | 103 | 147 | 954 | 597 | 197 | 279 | 114 | 41 | 10 |

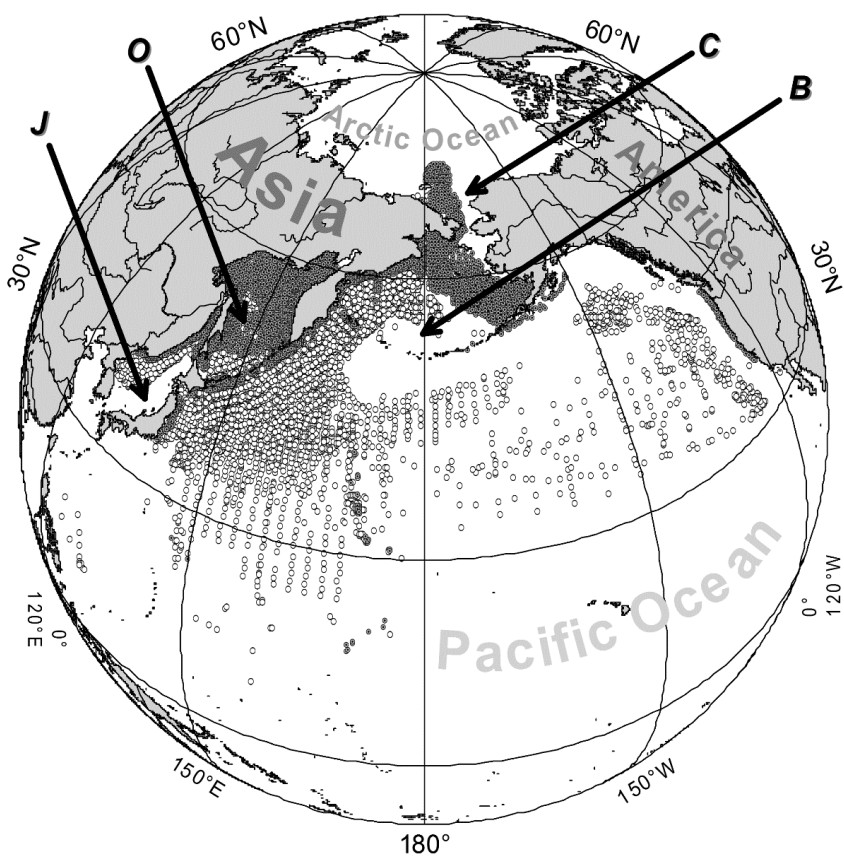

**Figure 1** Spatial distribution of midwater (open circles) and bottom (dark circles) trawl stations

used to compile the trawl macrofauna checklist. Letters indicate: C – Chukchi Sea, B – Bering

Sea, O – Sea of Okhotsk, and J – Sea of Japan (from Volvenko et al., 2018).





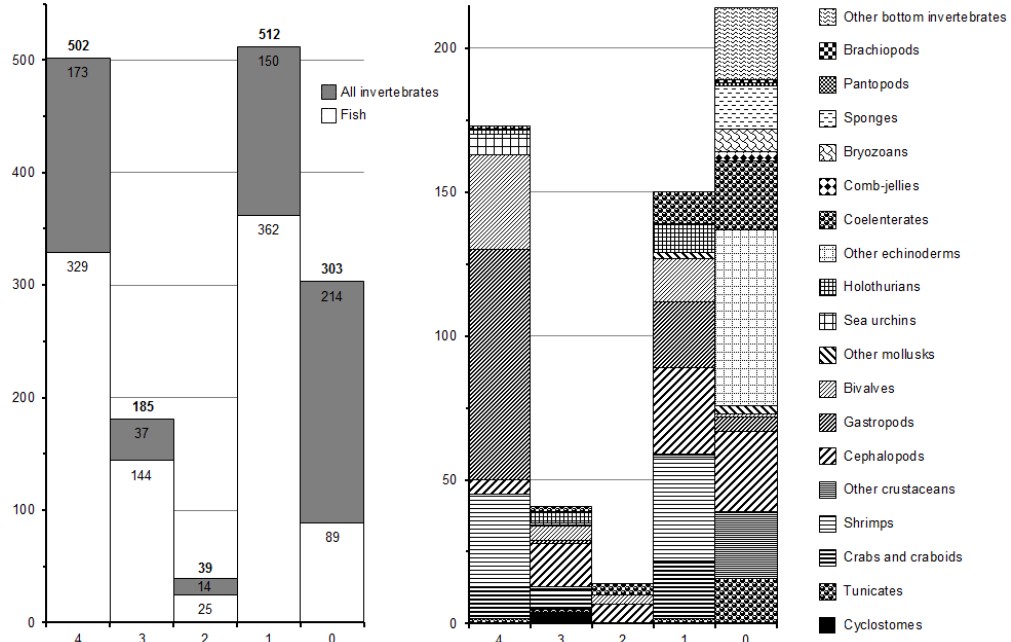

**Figure 2** Number of species from different taxa (Y-axis) in five commercial categories (X-axis).

On the left graph fish and invertebrates, on the right – all taxa except for fish.



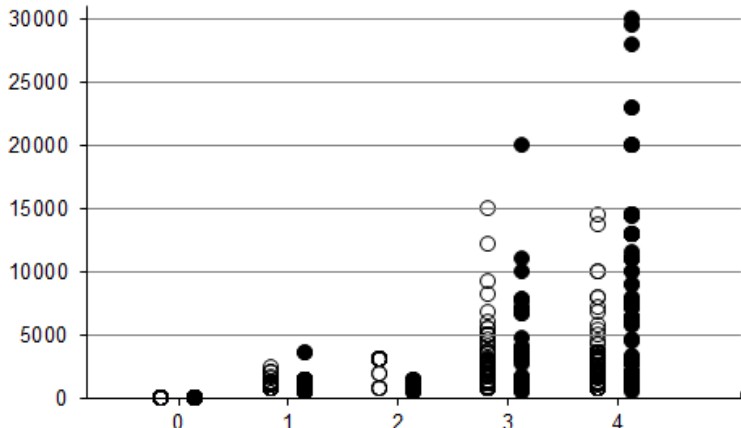

**Figure 3** Relationship between species commercial categories (0-4 on X-axis) and price of a

product from this species (USD per ton on Y-axis). Each circle – one of 1541 species from the

checklist (Supplementary Table): open circles – fish, dark circles – invertebrates.

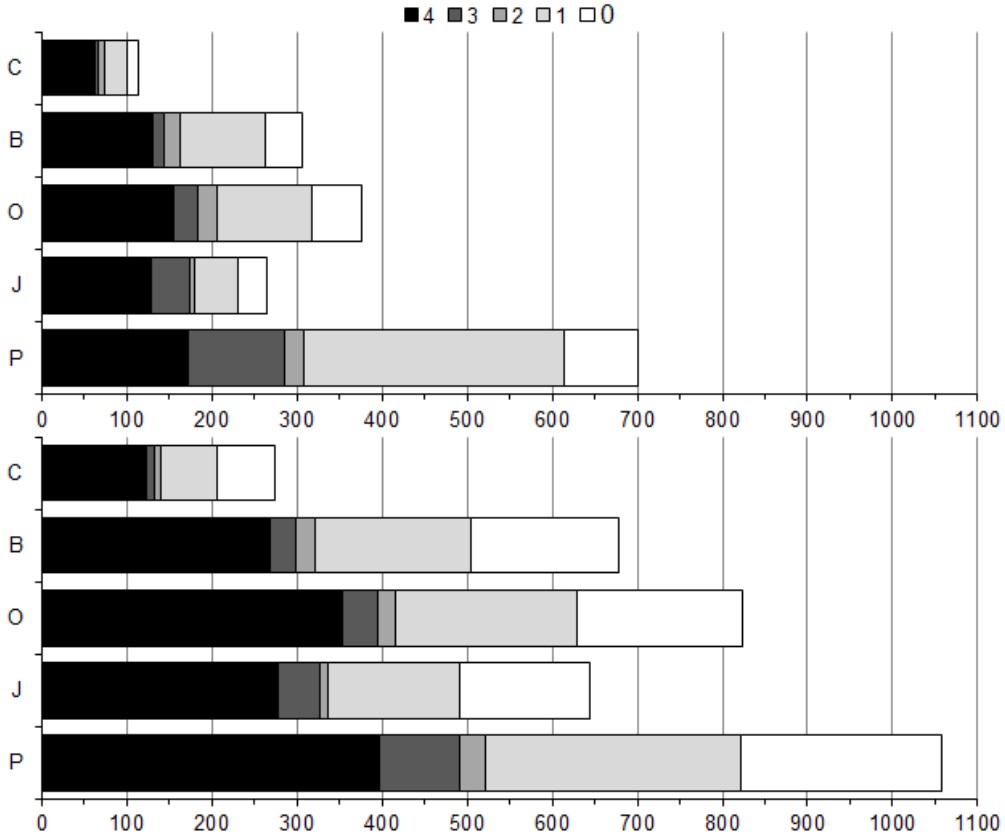

**Figure 4** Number of species in five commercial categories (4-0) in different basins: upper graph – pelagic zone; bottom graph – benthic zone. X-axis indicates number of species; Y-axis indicates basins: C – Chukchi Sea, B – Bering Sea, O – Sea of Okhotsk, J – Sea of Japan and P – Pacific Ocean. Categories are shown in different colour.



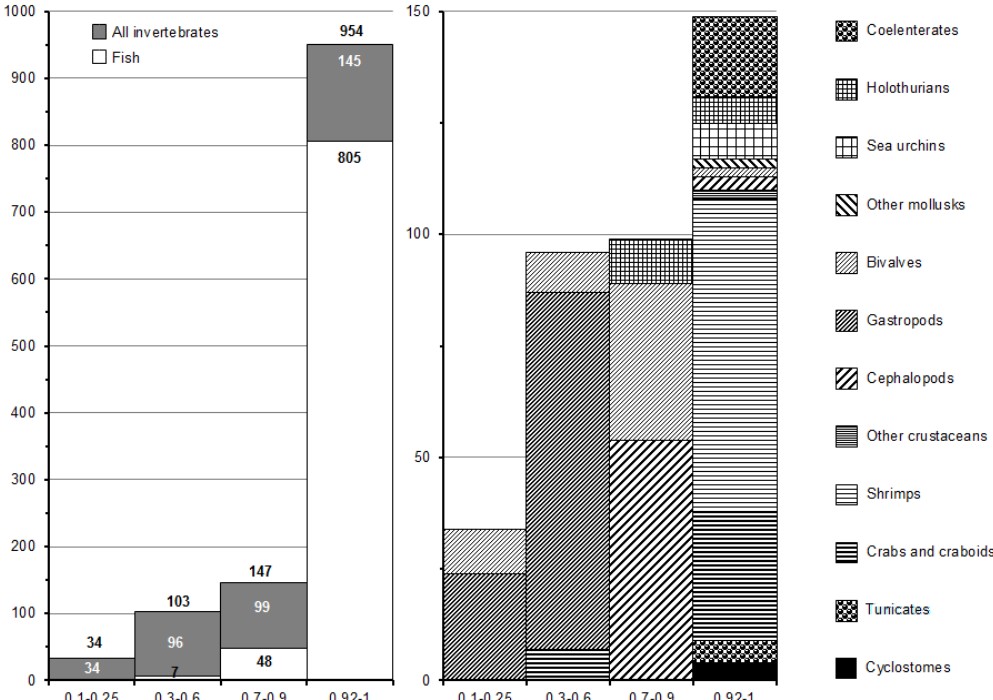

**Figure 5** Number of commercial species of different taxa in four technological groups. X-axis indicates product yield (percent from the raw weight). Y-axis – number of species. On the left graph fish and invertebrates, on the right – all taxa except for fish.

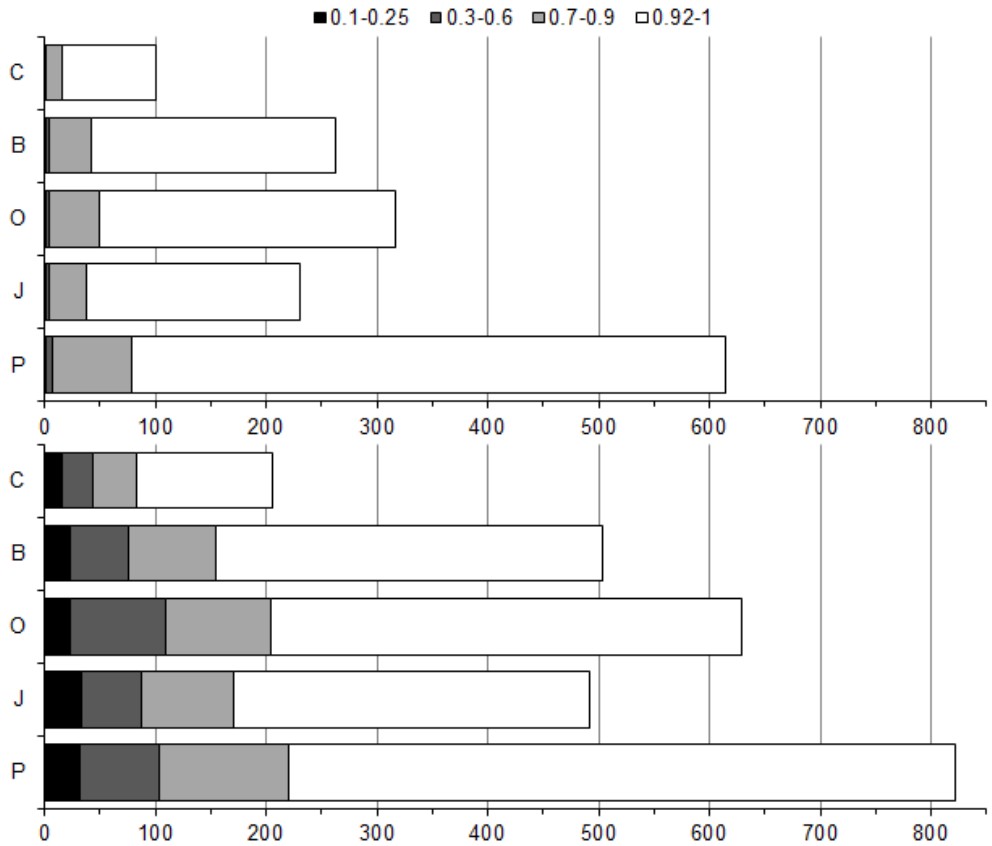

**Figure 6** Number of commercial species in four technological groups with different product

yield in different basins: upper graph – pelagic zone; bottom graph – benthic zone. X-axis

indicates number of species; Y-axis indicates basins: C – Chukchi Sea, B – Bering Sea, O – Sea

of Okhotsk, J – Sea of Japan and P – Pacific Ocean. Product yield values are shown in different

colour.



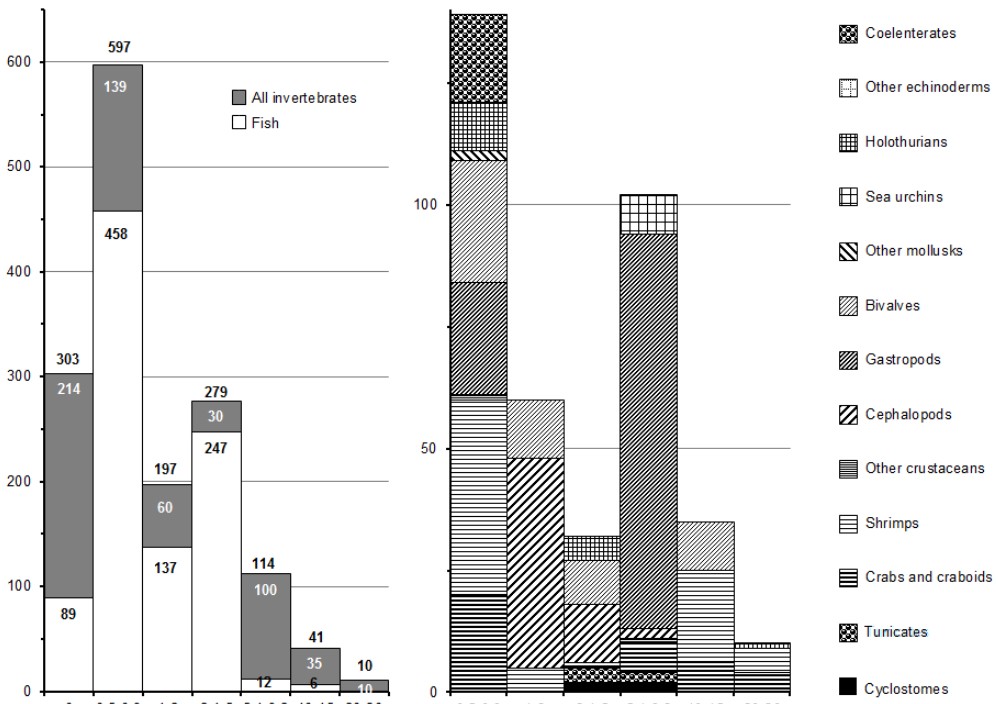

**Figure 7** Number of species in different price categories. X-axis indicates price category (thousand USD per ton), Y-axis indicates number of species. On the left graph fish and invertebrates, on the right – all taxa except for fish. Species without commercial value in price category 0 USD per ton are shown on the left graph and are not shown on the right graph.

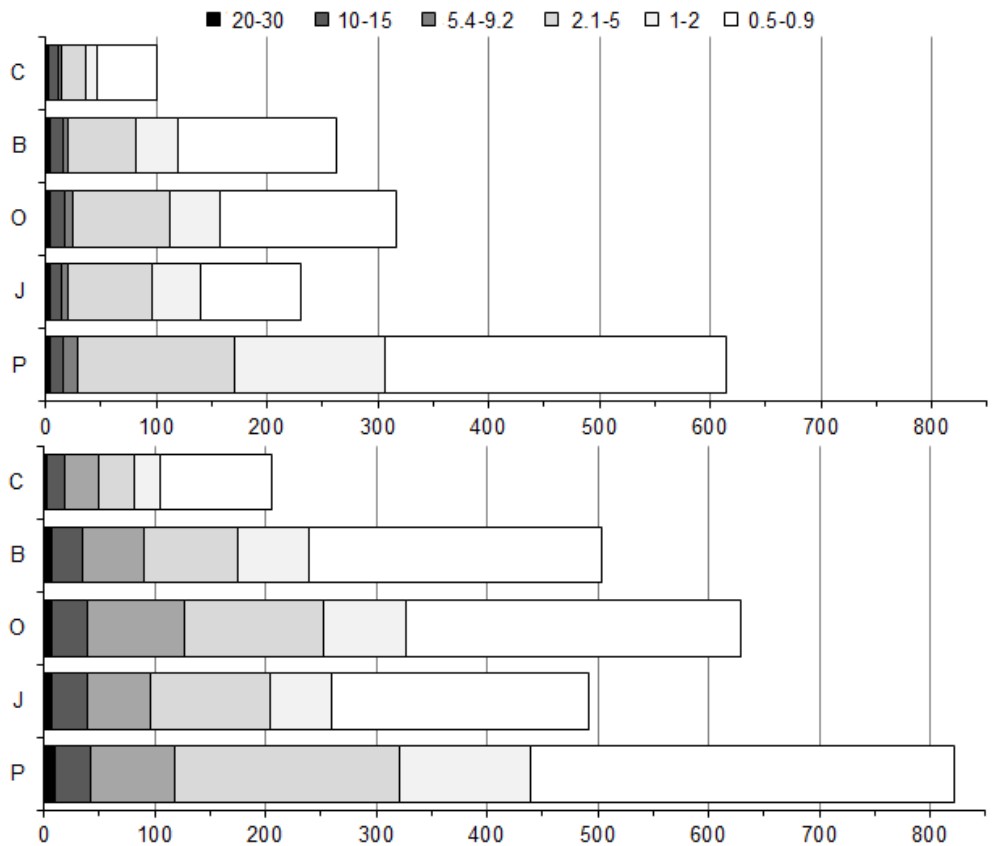

**Figure 8** Number of species from different price categories in different basins: upper graph –

pelagic zone; bottom graph – benthic zone. X-axis indicates number of species; Y-axis indicates

basin: C – Chukchi Sea, B – Bering Sea, O – Sea of Okhotsk, J – Sea of Japan and P – Pacific

Ocean. Price values (thousand USD per ton) are shown in different colour.

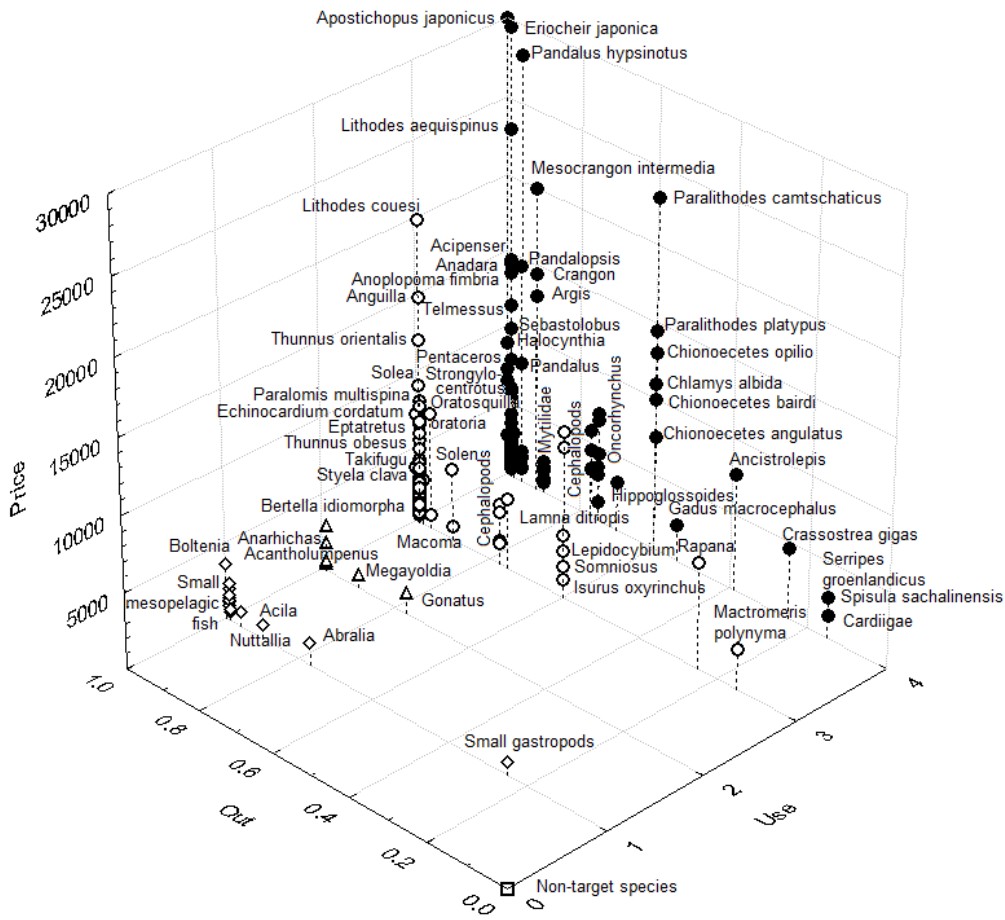

**Figure 9** Distribution of all macrofauna species from the study area by categories of commercial

importance (X-axis "Use", 0-4), product yield (Y-axis "Out", proportion of the raw weight) and

price (Z-axis "Price", USD per ton). The scientific names of standing out species are shown

where space allows. Most of the signs correspond to several species with similar features.