# Peer review of "A suggested short running title"

_Earth System Science Data, 2019_

## Referee Comment (RC1) · Michael Vecchione (Referee) · 30 Jul 2019

This review is unusual for me. Nominally, the manuscript, together with the data avail-abel from PANGAEA, is a data discussion rather than a presentation of research re-sults and inferrences. The data in question, a massive set of survey-trawl data, have been summarized by these authors recently in Nature/Scientific Reports. Whereas the previous summary focussed on occurrence and relative abundance, this manuscript develops inferences about potential commercial value. There is considerable overlap between this manuscript and the previous publication, including some short passages that are reproduced verbatim. I think the data set considered here is immensely im-

portant. However, in my opinion the primary importance is what was covered in the Scientific Reports publication. Whether there are ethical issues relative to advocacy of expanded exploitation of living marine resources in this region is a question that might properly be raised by readers. However, as a scientific reviewer, I will leave that question to the editors. If the goal here is "to analyse the importance of trawl macrofauna to fisheries", I think that importance should include more than current or potential monetary value. Importance should include ecological relationships such as predator/prey and habitat structure. A unfished species may be important to to the food web supporting fished species. Similarly a species with no commercial value may provide nursery habitat or protection from predators.

Specific comments l. 98 – A midwater tow to a depth of 2200 m is not mesopelagic. l. 104 – Re "known to occur", were these additional data only from trawl studies? If so, please state explicitly. ll. 106-107 – This sentence lacks a verb. l. 243 – "Coelenterates" is considered by some to include ctenophores (i.e., comb jellies in the following category). I think this it would be better for this category to be "Cnidarians". ll. 282-283 – Change "almost two times lower" to "approximately half of the respective numbers for fish species". l. 302 – "Shellfish" generally refers to crustaceans + molluscs. Therefore "crustaceans" are "shellfish". I think you mean "molluscs" rather than "shellfish". ll. 337, 342 (and elsewhere) – If "squid" refers to >1 species, it should be "squids". l. 384 – Change "shell" to "shelled". l. 395 (and elsewhere) – If "fish" and "shrimp" refer to >1 species, they should be "fishes" and "shrimps.

---

## Author Comment (AC1) · 2 Aug 2019

Dear Editors,

We would like to thank Michael Vecchione (Referee) for providing a very constructive review that has resulted in improvement of our manuscript. We are very pleased that the Referee called our data set "immensely important". Please find below a response to the Reviewer comments, which provides point-by-point answer to each of the referee's comments, and where we tried to address concisely each of the issues raised. (Reviewer's comments are in square brackets).

[Figure]

[This review is unusual for me. Nominally, the manuscript, together with the data available from PANGAEA, is a data discussion rather than a presentation of research results and inferences.]

Indeed, our review discusses the research results presented in PANGAEA, as well as methods for obtaining and processing those data, with inferences given in the Conclusions section at the end of the manuscript. We believe this is the usual scheme for treating data presented to the Earth System Science Data and it is consistent with the rules for authors.

[The data in question, a massive set of survey-trawl data, have been summarized by these authors recently in Nature/Scientific Reports. Whereas the previous summary focused on occurrence and relative abundance, this manuscript develops inferences about potential commercial value. There is considerable overlap between this manuscript and the previous publication, including some short passages that are reproduced verbatim.]

This work is indeed a continuation of the recently published study in the Scientific Reports, however, a lot of new data has been added to the new data set. In addition, the new manuscript has completely different goals, objectives, results and conclusions compared to the published paper. Some coincidences of the texts of these two articles (including verbatim) are found in the description of data collection methods, since these methods partially overlap. We made minimal repetitions so that the meaning of the second paper was understood without reading the first one. Although, where it is needed for an in-depth understanding of the techniques, the present manuscript provides links to the article in the Scientific Reports.

[I think the data set considered here is immensely important. However, in my opinion the primary importance is what was covered in the Scientific Reports publication.]

In the Scientific Reports publication, the focus was on some fundamental issues of ecology and biogeography. The new manuscript focuses more on applied issues related to fishing and rational exploitation of biological resources of the vast region, one of the most productive on Earth. We believe that this problem is also important, and similar publications on other sea areas would be useful for generating a comprehensive inventory of renewable resources of the entire World Oceans. We also draw your attention to the fact that the species list published in PANGAEA can be used to assess the economic value of biological resources or damages to marine ecosystems resulting from anthropogenic impact, including pollution, hydro-technical constructions, oil and gas extraction, tanker or nuclear reactors accidents, etc. The list published earlier in the Scientific Reports is not suitable for such purposes.

[Whether there are ethical issues relative to advocacy of expanded exploitation of living marine resources in this region is a question that might properly be raised by readers. However, as a scientific reviewer, I will leave that question to the editors.]

As for "ethical issues relative to advocacy of expanded exploitation of living marine resources in this region", we would like to note that in our study we do not call for strengthening this exploitation, but rather on the contrary – for making it more rational, while expanding the range of fishery products, transfer part of the load from traditionally most intensively exploited species to other species – currently underutilized and unused.

[If the goal here is "to analyse the importance of trawl macrofauna to fisheries", I think that importance should include more than current or potential monetary value. Importance should include ecological relationships such as predator/prey and habitat structure. An unfished species may be important to the food web supporting fished species. Similarly, a species with no commercial value may provide nursery habitat or protection from predators.]

Questions about "ecological relationships such as predator / prey and habitat structure", like many others, will be the subject of our future publications, since it is not possible to consider all these problems in one manuscript. To conduct a complete

analysis of all aspects of the use of trawl macrofauna in fishing, a series of large-scale review publications is needed.

[Specific comments]

We are very grateful to the Referee for specific comments on the text of the manuscript on the following lines (l.). Our mistakes and typos in these places are corrected in a new version of the manuscript:

l. 98 – "mesopelagic" is replaced by "bathypelagic"; l. 105 – "known to occur" is replaced by "known to occur (not only from trawl studies)"; l. 108 – the verb "are" is added; l. 244 – "Coelenterates" is replaced by "Cnidarians"; ll. 283-284 – "almost two times lower" is replaced by "approximately half of the respective numbers for fish species"; l. 305 – "shellfish" is replaced by "molluscs"; ll. 340, 345, 443 – "squid" is replaced by "squids"; l. 387 – "shell" is replaced by "shelled"; ll. 32, 294, 398, 490, 504, 513, 531, 536, 566 – "fish" is replaced by "fishes"; l. 398 – "shrimp" is replaced by "shrimps"; ll. 306, 332, 449 – "jellyfish" is replaced by "jellyfishes".

Since every review comment was highly appropriate and valuable, we have followed recommendations of the Reviewer as much as possible. The revised manuscript version was prepared for uploaded to the website.

Best regards,

I. Volvenko, A. Orlov, A. Gebruk, O. Katugin, A. Ogorodnikova, G. Vinogradov, O. Maznikova
* * *

---

## Referee Comment (RC2) · Anonymous Referee #2 · 10 Aug 2019

Dear editor and authors,

I find it very difficult to understand the rational behind the current presentation of the manuscript. In general, the idea to link data from scientific surveys to catch data and potential value is not overly unique and it is very difficult for me to understand who would use such a database...

I have several general comments and concerns: 1) the data set is covering the period from 1977-2014 and therefore also covers a period with significant environmental changes. Thus it would be very interesting to understand which species have been found in which area at which time and not only present a checklist with information

wheteher a soecies has been caught with which gear... I would imagine that coverage of surveys in a given year varies and thus a statsitical survey index for each species for each year would be deemed more useful than just the presence or absence in a given area... Also, how is abundance and distribution related to a number of variable like water temperature or atmospheric indices influencing temperature and currents? Can future abundance and distribution be estimated?

2) the same is true for catch data: instead of a current status of exploitation indicating just if a species is exploited or not, the actual catch in a given year would be a useful information to identify trends in exploitation

3) ex-vessel prices are dependent on many things and are normally highly volatile. Thus a minimum price seems not very meaningful as information. Again, the trend of the price in relation to at least the catch in that specific area and the global catch of that species would be more helpful. If there are several global stocks exploited of the same species, e.g. cod. then price of fish harvested from small stocks is likely highly dependent on the largest catch of that species (e.g. Barents Sea cod is driving the global price and not North Sea cod or Baltic cod...). Price of a given species is also dependent on current preferences and substitution elasticity, i.e. that if species are easy to substitute with other fish like Alaska Pollock with other fish with white meat, than price is not only dependent on the current catch of this one species. The influence of preference can be clearly seen for example on roe of pacific herring, which is mainly consumed as sashimi and sushi in Japan. The change in preference over the last decade has already a large impact on demand, price and catch...

A very valuable information would be the relation between harvested 'species' and actual species, e.g. are several species harvested as one species than there is the risk of overharvesting the least resilient species of that complex.

In light of my above comments I am left a bit clueless of what to recommend, but as this is an interactive review, I am very interested in the answer of the authors on this.

---

## Author Comment (AC2) · 13 Aug 2019

Dear Editors,

We would like to thank the Anonymous Referee #2 for important comments. We see them in general as recommendations for future extension of our present work. Please find below point-by-point answers to each of the Referee's comments.

Let us start with the question who and how could use the database (species checklist) that we have collected and presented in PANGAEA. First of all, this is a catalogue that can be used by fishermen, fishery managers, officials and policy makers for the de-

[Figure]

velopment of bioresource management, aquaculture and conservation, assessment of environmental damage caused by anthropogenic impact. Secondly, we have already made the first analysis of this checklist in our review and came up with six practically important conclusions, which were formulated at the end of the manuscript. We gave in it a specific example of publication [1] in which an approximate economic assessment of biological resources of the Sea of Okhotsk was made according to preliminary data. At present this estimate can be refined using our data and extended for other Far Eastern seas (hereinafter, by this term we mean the Sea of Japan, the Sea of Okhotsk, the Bering Sea and the Chukchi Sea.). We could also mention similar economic assessment [2] with the analysis of rationality and efficiency of the use of biological resources of the western part of the Bering Sea. However, unfortunately, it has been published only in Russian and was not translated into English. In the near future, we plan a series of similar publications in English on all Far Eastern seas and the North Pacific Ocean. It will be based on the checklist presented in PANGAEA, combined with the data of scientific trawl surveys and official data on fishing. Anyone interested could do this based on our checklist and catch data from FAO directories (e.g. [3-9]) or from the website of the Russian Federal Fisheries Agency (available at http://fish.gov.ru/otraslevaya-deyatelnost/ekonomika-otrasli/statistika-i-analitika) or the Center of Fishery Monitoring and Communications (http://cfmc.ru), as well as atlases [10-13] and tabular guides [14-25] on pelagic and bottom macro fauna for each of the Far Eastern sea, that have been published earlier. However, before that we would like to supplement the databases on trawl catches [26-28] with materials from the latest research surveys, including those conducted in 2018. Also, we already mentioned the importance of the checklist published in PANGAEA for assessing environmental damage to marine ecosystems resulting from anthropogenic impacts, including pollution, hydraulic structures, oil and gas production, accidents in tankers or nuclear reactors, etc. In the simplest case, in order to estimate such damage in terms of cost, total destructed biomass should be multiplied by possible product yield and prices for respective species shown in the present study. For more comprehensive assessment,
the same procedure should be conducted considering the potential offspring of those organisms over a certain period of time, and resulting amounts should be summed up. Environmental and regulatory authorities use such assessments to recover damages.

Comments on the three general concerns of the Referee.

1) The data set indeed covers the period from 1977-2014 and therefore also covers a period with significant environmental changes. It is easy to see which species have been found in which area at which time based on atlases [10-13] and table directories [14-25], as well as on many publications on this topic. Also, there are publications on how abundance and distribution are related to a number of variables, like water temperature or atmospheric indices influencing temperature and currents (e.g. some of the latest publications on these topics on the Sea of Okhotsk [29-31]). The checklist presented in PANGAEA is intended for other purposes – to translate biological (ecologic, biogeographic) assessments into economic assessments.

2) It is clear that the actual catch in a given year would be a useful information to identify trends in exploitation and provide links to open sources of such information. We also note that this information is not available for all species listed in the checklist, but only for the most abundant. Moreover, we plan to analyze all that information in our next publications presenting it in monetary terms using our checklist.

3) The Referee correctly noted that the state of the fish market depends not only on changes in the abundance of one or another species in nature, but also on price dynamics, which in turn depends on many factors and, as a rule, is very variable. We also point out this in our review and name some of these factors. The minimum prices are useful to demonstrate that the value of available biological resources at certain place at one time is no less than a certain amount in dollars. In other words, it shows the minimum value of damage to nature. In the future, we planned to supplement our checklist with maximum and average prices, but not at this stage.

We also share the Referee's concern that several species harvested as one species

carry the risk of overharvesting the least resilient species of that assemblage (see e.g. [32]). However, it is not easy to cope with that problem. In their official catch reports, fishermen often do not specify, e.g., squids, sharks or gobies by species. Fishermen do not believe that they are obliged to do this, and fishing rules do not prohibit this approach. Total and accurate species identification of catches is possible only in scientific surveys, the data from which we have used to create our checklist published in PANGAEA. Since each comment of the Referee was very relevant and valuable, in our future work, we will try to follow most of his recommendations.

References

[1] Lukyanova, O.N., Volvenko, I.V., Ogorodnikova, A.A. and Anferova, E.N.: The economic valuation of biological resources and ecosystem services in the Sea of Okhotsk, Russian Journal of Marine Biology, 42, 602–607, https://link.springer.com/article/10.1134/S1063074016070075, 2016. [2] Volvenko, I.V.: Dataware support of comprehensive studies of Northwestern Pacific aquatic biological resources. Part 3. GIS, atlases, reference books, further prospects of the concept, Trudy VNIRO, 157, 100-126, http://vniro.ru/files/trydi_vniro/archive/tv_2015_t_157_article_6.pdf, 2015. [3] FAO yearbook. Fishery and Aquaculture Statistics, Rome, Food and Agriculture Organization of the United Nations, http://www.fao.org/docrep/015/ba0058t/ba0058t.pdf, 2010. [4] FAO yearbook. Fishery and Aquaculture Statistics, Rome, Food and Agriculture Organization of the United Nations, http://www.fao.org/3/a-i3740t.pdf, 2012. [5] FAO yearbook. Fishery and Aquaculture Statistics, Rome, Food and Agriculture Organization of the United Nations. http://www.fao.org/3/a-i5716t.pdf, 2014. [6] The state of world fisheries and aquaculture, Rome, Food and Agriculture Organization of the United Nations, http://www.fao.org/3/a-y7300e.pdf, 2002. [7] The state of world fisheries and aquaculture, Rome, Food and Agriculture Organization of the United Nations, http://www.fao.org/docrep/016/i2727e/i2727e.pdf, 2012. [8] The state of world fisheries and aquaculture, Rome, Food and Agriculture Organization of the United Nations, http://www.fao.org/3/a-i3720e.pdf, 2014. [9] The state of world fisheries and aquaculture, Rome, Food and Agriculture Organization of the United Nations, http://www.fao.org/3/a-i5555e.pdf, 2016. [10] Shuntov, V.P. and Bocharov, L.N., Eds.: Atlas of Quantitative Distribution of Nekton Species in the Okhotsk Sea. Moscow: National Fish Resources, https://www.researchgate.net/publication/259297238_Atlas_kolicestvennogo_raspredelenia_nektona_v_Ohotskom_more, 2003. [11] Shuntov ,V.P. and Bocharov, L.N., Eds.: Atlas of Quantitative Distribution of Nekton Species in the Northwestern Part of the Japan/East Sea. Moscow: National Fish Resources, https://www.researchgate.net/publication/281976394_Atlas_of_nekton_species_quantitative_distribution_in_the_north-western_part_of_the_JapanEast_Sea, 2004. [12] Shuntov, V.P. and Bocharov, L.N., Eds.: Atlas of Quantitative Distribution of Nekton Species in the Northwestern Part of the Pacific Ocean. Moscow: National Fish Resources, https://www.researchgate.net/publication/259297325_Atlas_kolicestvennogo_raspredelenia_nektona_v_severo-zapadnoj_casti_Tihogo_okeana, 2005. [13] Shuntov, V.P and Bocharov, L.N., Eds.: Atlas of Quantitative Distribution of Nekton Species in the Western Part of the Bering Sea. Moscow: National Fish Resources, https://www.researchgate.net/publication/268689118_Atlas_kolicestvennogo_raspredelenia_nektona_v_zapadnoj_casti_B 2006. [14] Shuntov, V.P. and Bocharov, L.N., Eds.: Nekton of the Okhotsk Sea. Tables of Abundance, Biomass and Species Ratio. Vladivostok: TINRO-Center, https://www.researchgate.net/publication/282059498_Nekton_of_the_Okhotsk_Sea_Abundance_biomass_and_species_r 2003. [15] Shuntov, V.P. and Bocharov, L.N., Eds.: Nekton of the Northwestern Part of the Japan/East Sea. Tables of Abundance, Biomass and Species Ratio. Vladivostok: TINRO-Center, https://www.researchgate.net/publication/259297240_Nekton_severo-zapadnoj_casti_Aponskogo_mora_Tablicy_cislennosti_biomassy_i_sootnosenia_vidov, 2004. [16] Shuntov, V.P. and Bocharov, L.N., Eds.: Nekton of the Northwestern Part of the Pacific Ocean. Tables of Abundance, Biomass and Species Ratio. Vladivostok: TINRO-Center, https://www.researchgate.net/publication/259297241_Nekton_severo-zapadnoj_casti_Tihogo_okeana_Tablicy_cislennosti_biomassy_i_sootnosenia_vidov,

2005. [17] Shuntov, V.P and Bocharov, L.N., Eds.: Nekton of the Western Part of the Bering Sea. Tables of Abundance, Biomass and Species Ratio. Vladivostok: TINRO-Center, https://www.researchgate.net/publication/282060996_Nekton_of_the_western_part_of_the_Bering_Sea_Abundance_bior 2006. [18] Shuntov, V.P. and Bocharov, L.N., Eds.: Pelagic Macrofauna of the Northwestern Part of the Pacific Ocean: Tables of Occurrence, Abundance and Biomass. 1979-2009. Vladivostok: TINRO-Center, https://www.researchgate.net/publication/282135949_Pelagic_macrofauna_of_the_northwestern_Pacific_occurrence_abu 2009, 2012. [19] Shuntov, V.P. and Bocharov, L.N., Eds.: Pelagic Macrofauna of the Okhotsk Sea: Tables of Occurrence, Abundance and Biomass. 1984-2009. Vladivostok: TINRO-Center, https://www.researchgate.net/publication/282135944_Pelagic_macrofauna_of_the_Okhotsk_Sea_occurrence_abundance 2009, 2012. [20] Shuntov, V.P. and Bocharov, L.N., Eds.: Pelagic Macrofauna of the Western Part of the Bering Sea: Tables of Occurrence, Abundance and Biomass. 1982-2009. Vladivostok: TINRO-Center, https://www.researchgate.net/publication/282135940_Pelagic_macrofauna_of_the_western_part_of_the_Bering_Sea_oc 2009, 2012. [21] Shuntov, V.P. and Bocharov, L.N., Eds.: Benthic Macrofauna of the Northwestern Part of the Japan/East Sea: Tables of Occurrence, Abundance and Biomass. 1978-2010. Vladivostok: TINRO-Center, https://www.researchgate.net/publication/281976537_Benthic_macrofauna_of_the_the_northwestern_part_of_Japan_Eas 2010, 2014. [22] Shuntov, V.P. and Bocharov, L.N., Eds.: Benthic Macrofauna of the Northwestern Part of the Pacific Ocean: Tables of Occurrence, Abundance and Biomass. 1977-2008. Vladivostok: TINRO-Center, https://www.researchgate.net/publication/282057693_Benthic_macrofauna_of_the_northwestern_Pacific_occurrence_abu 2008, 2014. [23] Shuntov, V.P. and Bocharov, L.N., Eds.: Benthic Macrofauna of the Peter the Great Bay (Japan/East Sea): Tables of Occurrence, Abundance and Biomass. 1978-2009. Vladivostok: TINRO-Center, https://www.researchgate.net/publication/282057904_Benthic_macrofauna_of_Peter_the_Great_Bay_JapanEast_Sea_oc 2009, 2014. [24] Shuntov, V.P. and Bocharov, L.N., Eds.: Benthic Macrofauna of the Sea of Okhotsk: Tables of Occurrence, Abundance and Biomass. 1977-2010. Vladivostok: TINRO-Center, https://www.researchgate.net/publication/268632096_Makrofauna_bentali_Ohotskogo_mora_tablicy_vstrecaemosti_cislen 2010_Benthic_macrofauna_of_the_Okhotsk_Sea_occurrence_abundance_and_biomass_1977-2010, 2014. [25] Shuntov, V.P. and Bocharov, L.N., Eds.: Benthic Macrofauna of the Western Part of the Bering Sea: Tables of Occurrence, Abundance and Biomass. 1977-2010. Vladivostok: TINRO-Center, https://www.researchgate.net/publication/268632126_Makrofauna_bentali_zapadnoj_casti_Beringova_mora_tablicy_vstre 2010_Benthic_macrofauna_of_the_western_part_of_the_Bering_Sea_occurrence_abundance_and_biomass_1977-2, 2014. [26] Volvenko, I.V.: The new large database of the Russian bottom trawl surveys in the Far Eastern seas and the North Pacific Ocean in 1977-2010, International Journal of Environmental Monitoring and Analysis, 2, 302-312, http://www.sciencepublishinggroup.com/journal/paperinfo?journalid=162&doi=10.11648/j.ijema.20140206.12; http://dx.doi.org/10.11648/j.ijema.20140206.12, 2014. [27] Volvenko, I.V.: The concept of information support for bioresource and ecosystem research in the North-West Pacific: theory and practical implementation, Natural Resources, 7, 40-50, http://www.scirp.org/JOURNAL/PaperInformation.aspx?PaperID=62909, 2016. [28] Volvenko, I.V. and Kulik, V.V.: Updated and extended database of the pelagic trawl surveys in the Far Eastern seas and North Pacific Ocean in 1979-2009, Russian Journal of Marine Biology, 37, 513-532, http://dx.doi.org/10.1134/S1063074011070078, 2011. [29] Zuenko Y.I., Aseeva N.L., Glebova S.Y., Gostrenko L.M., Dubinina A.Y., Dulepova E.P., Zolotov A.O., Loboda S.V., Lysenko A.V., Matveev V.I., Muktepavel L.S., Ovsyannikov E.E., Figurkin A.L., Shatilina T.A.: Recent changes in the Okhotsk sea ecosystem (2008–2018), Izvestiya TINRO, 197, 35-61, https://doi.org/10.26428/1606-9919-2019-197-35-61, 2019. [30] Shuntov V.P., Ivanov O.A., Gorbatenko K.M.: What happened in the ecosystem of the Okhotsk sea in 2008–2018?, Izvestiya TINRO.;197:62-82. (In Russ.) https://doi.org/10.26428/1606-9919-2019-197-62-82, 2019. [31] Shuntov V.P., Ivanov O.A., Dulepova E.P.: Biological resources in the Sea of Okhotsk Large Marine Ecosystem: Theirstatus and commercial use, Deep-Sea

Research Part II, 163, 33-45, https://doi.org/10.1016/j.dsr2.2019.01.006, 2019. [32] Volvenko I.V.: The importance of species diversity and its components as criteria for selecting nature conservation areas, Russian Journal of Marine Biology, 37, 604-607, https://link.springer.com/content/pdf/10.1134%2FS106307401107008X.pdf, 2011.

Best regards,

I. Volvenko, A. Orlov, A. Gebruk, O. Katugin, A. Ogorodnikova, G. Vinogradov, O. Maznikova

---

## Editor Comment (EC1) · Dirk Fleischer (Editor) · 30 Aug 2019

Dear Authors,

I do agree with the reviewers that the manuscript is worth publishing and the data set contains valuable information, but I also do agree with the review comments, that the aggregated nature of the data set does not support broad future reusability of the provided information. From the general list it is not possible to derive if the geographical distribution over this long time series has changed. There is also no information on the annual coverage of the trawl surveys in the manuscript. Again the aggregation in the provided list does not indicate if a species occurred just recently or already vanished

30 years ago. This is all very important for future use of the dataset and I can only recommend that you should provide the raw trawl data which I believe will be used by several other scientists. I recommend to publish directly the full data matrix, but I can understand if you prefer annual - station - presence/absence lists, which would be the step in the right direction.

I hope you consider this comment valuable, because the additional value added to your manuscript would most likely result in higher reuse rate and therefore citations of your manuscript

Dirk Fleischer

---

## Author Comment (AC3) · 6 Sep 2019

Dear Dirk Fleischer,

We are grateful for the opportunity to resubmit a revised version of our paper "Trawl macrofauna of the Far-Eastern Seas and North Pacific: proportion of commercial species, potential product yield, and price range", and would like to thank you for your assistance in the submission and interactive discussion processes. We are very pleased that you and reviewers find our "manuscript is worth publishing and the data set contains valuable information".

[Figure]

We would also like to thank the reviewers for providing many constructive comments, which are very valuable in improving the quality of the manuscript and recommendations for future extension of our present work. All changes made to the original version are marked with yellow in revised version, and are listed below together with the line number:

l. 98 - "mesopelagic" is replaced by "bathypelagic"; l. 105 - "known to occur" is replaced by "known to occur (not only from trawl studies)"; l. 108 - the verb "are" is added; l. 244 - "Coelenterates" is replaced by "Cnidarians"; ll. 283-284 - "almost two times lower" is replaced by "approximately half of the respective numbers for fish species"; l. 305 - "shellfish" is replaced by "molluscs"; ll. 340, 345, 443 - "squid" is replaced by "squids"; l. 387 - "shell" is replaced by "shelled"; ll. 32, 294, 398, 490, 504, 513, 531, 536, 566 - "fish" is replaced by "fishes"; l. 398 - "shrimp" is replaced by "shrimps"; ll. 306, 332, 449 - "jellyfish" is replaced by "jellyfishes", ll. 58, 59, 60 - page number changes due to the above corrections.

Next, we would like to answer your more general comments.

In the manuscript, indeed no information is given on the annual coverage of trawl surveys. If you insist, we can add another table to the Materials and Methods section. In this table, the rows will correspond to years, the columns to basins, and contents of the cells to the number of trawl samples in given locality for each year. However, we believe that such a large table will take up a lot of extra space without adding useful information, since this paper does not consider the long-term variability of species richness and/or species composition in the surveyed region.

We plan to consider changes in the number and geographical distribution of species among seasons and perennial periods in our next publications because this is a separate topic that goes beyond the stated goals and objectives of the present article. It is not possible to consider all problems in one manuscript, a series of large-scale publications is needed for this. We do believe that without the time-scale analysis, the

information we provide in our manuscript is still important and useful.

In conclusion, you recommend us to "provide the raw trawl data". Unfortunately, we cannot do this because we are not the owners of this data. The Pacific Branch of Russian Federal Research Institute of Fisheries and Oceanography (TINRO) provides us with the opportunity to use these data and publish the results obtained on them, but it strictly prohibits transferring the source data to third parties as well as publishing the raw trawl data.

Best regards,

I. Volvenko, A. Orlov, A. Gebruk, O. Katugin, A. Ogorodnikova, G. Vinogradov, O. Maznikova